# Optimization of rice panicle architecture by specifically suppressing ligand–receptor pairs

Tao Guo [1,2,6], Zi-Qi Lu[1,2,3,6], Yehui Xiong[4,6], Jun-Xiang Shan [1,2], Wang-Wei Ye[1,2], Nai-Qian Dong [1,2], Yi Kan[1,2], Yi-Bing Yang[1,2,5], Huai-Yu Zhao[1,2,5], Hong-Xiao Yu[1,2,5], Shuang-Qin Guo[1,2,5], Jie-Jie Lei[1,2,5], Ben Liao[1,2,3], Jijie Chai [4] ✉ & Hong-Xuan Lin [1,2,3,5] ✉

Rice panicle architecture determines the grain number per panicle and therefore impacts grain yield. The OsER1–OsMKKK10–OsMKK4–OsMPK6 pathway shapes panicle architecture by regulating cytokinin metabolism. However, the specific upstream ligands perceived by the OsER1 receptor are unknown. Here, we report that the EPIDERMAL PATTERNING FACTOR (EPF)/ EPF-LIKE (EPFL) small secreted peptide family members OsEPFL6, OsEPFL7, OsEPFL8, and OsEPFL9 synergistically contribute to rice panicle morphogenesis by recognizing the OsER1 receptor and activating the mitogen-activated protein kinase cascade. Notably, *OsEPFL6*, *OsEPFL7*, *OsEPFL8*, and *OsEPFL9* negatively regulate spikelet number per panicle, but *OsEPFL8* also controls rice spikelet fertility. A *osepfl6 osepfl7 osepfl9* triple mutant had significantly enhanced grain yield without affecting spikelet fertility, suggesting that specifically suppressing the OsEPFL6–OsER1, OsEPFL7–OsER1, and OsEPFL9–OsER1 ligand–receptor pairs can optimize rice panicle architecture. These findings provide a framework for fundamental understanding of the role of ligand–receptor signaling in rice panicle development and demonstrate a potential method to overcome the trade-off between spikelet number and fertility.

Rice (*Oryza sativa*) serves as a staple cereal crop worldwide, feeding more than half of the global population. Crop yield is a complex trait that in rice depends on tiller number, grain weight, and grain number per panicle. Of these parameters, grain number per panicle is a more flexible determinant; it is composed of the primary and secondary panicle branches and the number of spikelets, and thus plays an important role in improving grain yield[1]. Shaping rice inflorescence architecture is a spatiotemporally programmed and interactive cellular process, in which inflorescence meristems differentiate into primary branch meristems attached to a central rachis[2]. The primary branch meristem is ultimately converted to a terminal spikelet meristem after forming several secondary branch meristems, with which the spikelet meristems are directly initiated on the primary and secondary branches, determining the final grain number per panicle. These spikelets develop terminal floral organs with palea and lemma, which determine grain size[2,3]. In rice, the molecular mechanism coordinating the trade-

[1]National Key Laboratory of Plant Molecular Genetics, CAS Centre for Excellence in Molecular Plant Sciences, Shanghai Institute of Plant Physiology and Ecology, Chinese Academy of Sciences, Shanghai 200032, China. [2]Guangdong Laboratory for Lingnan Modern Agriculture, Guangzhou 510642, China. [3]School of Life Science and Technology, ShanghaiTech University, Shanghai 201210, China. [4]Tsinghua-Peking Center for Life Sciences, School of Life Sciences, Tsinghua University, Beijing 100084, China. [5]University of the Chinese Academy of Sciences, Beijing 100049, China. [6]These authors contributed equally: Tao Guo, Zi-Qi Lu, Yehui Xiong. ✉e-mail: chaijj@tsinghua.edu.cn; hxlin@cemps.ac.cn

off between grain number per panicle and grain size has been uncovered[3]. However, a mechanism for effectively overcoming trade-offs among complex traits to improve rice yield by molecular design has been a challenge.

During rice panicle morphogenesis, the activities of different inflorescence meristem types determine functional tissue patterning. This process relies on cell–cell communication to specify cell fate and to coordinate developmental and environmental responses. Cumulative evidence has demonstrated that small secreted peptide (SSP) signaling plays an essential role in such intercellular communications and orchestrates various important biological processes in plants[4,5]. In *Arabidopsis thaliana*, the *CLAVATA* (*CLV*)–*WUSCHEL* (*WUS*) feedback loop specifies the stem cell niche within the shoot apical and floral meristems. *CLV1* encodes an extracellular leucine-rich repeat receptor-like kinase; *CLV3* encodes a CLV3/ESR-related (CLE) peptide that restricts stem cell proliferation and promotes differentiation[6–9]. In rice, mutants of the *CLV1* ortholog *FLORAL ORGAN NUMBER1* (*FON1*) and the *CLV3* ortholog *FON4* exhibit enlargement of the inflorescence meristem and an increase in the number of floral organs, consistent with *Arabidopsis clv* mutants[10–13]. This suggests that SSPs play conserved and pivotal roles in plant inflorescence development. Intriguingly, an EPIDERMAL PATTERNING FACTOR (EPF)/EPF-LIKE (EPFL) peptide family member, REGULATOR OF AWN ELONGATION2 (RAE2) or GRAIN NUMBER, GRAIN LENGTH AND AWN DEVELOPMENT1 (GAD1), has been shown to regulate awn development during rice domestication[14,15]. Moreover, numerous SSP family members and elicitors have been determined by omics-based screening to be induced by rice blast fungus, implying that SSPs are involved in the rice immune response[16]. Nevertheless, the small peptide ligands and targeted signaling receptors that are responsible for inflorescence meristem activity in rice remain largely unknown.

Emerging evidence has suggested that the ERECTA1 (OsER1)–OsMKKK10–OsMKK4–OsMPK6 pathway controls rice panicle morphogenesis by regulating cytokinin metabolism with the DROUGHT AND SALT TOLERANCE (DST)–CYTOKININ OXIDASE2 (OsCKX2) module[17,18], shedding light on how developmental signals maintain cytokinin homeostasis to shape plant inflorescence architecture. These findings raise the question of which kinds of SSPs responsible for rice panicle morphogenesis can recognize the OsER1 receptor. In *Arabidopsis*, together with the receptor-like protein TOO MANY MOUTHS (TMM), the ERECTA family (ERf) receptors can perceive the small peptide ligands EPF1 and EPF2 (which are secreted from neighboring stomatal precursors) to specify stomatal development[19,20]. However, another small peptide ligand (EPFL9/Stomagen) can compete with EPF1 and EPF2 for binding to the ERf–TMM complex[20–22]. Interestingly, EPFL4 and EPFL6, which are expressed in the endodermis and modulate elongation of inflorescence stems and vascular development, are recognized by a single ERf member without requiring TMM[20,23–25]. Moreover, the genetic analysis suggested that the ERf could sense four ligands in the shoot apical meristem (SAM) including the EPFL1, EPFL2, EPFL4, and EPFL6, which contribute to the establishment of meristem size and promotion of leaf initiation by expressing in the boundary region of the embryonic and vegetative SAM (EPFL1 and EPFL2), and at the periphery of the vegetative SAM (EPFL4 and EPFL6)[26]. This implies that the conserved EPF/EPFL family of SSPs could act as ligands of ERf. The specific small peptide ligands perceived by OsER1 are poorly understood in rice. A recent study revealed that *OsER1* is required for rice panicle morphogenesis and negatively regulates spikelet number per panicle[17]. Although the *oser1* mutant had a clear increase in spikelet number per panicle, the grain setting percentage was compromised; this suggested that *OsER1* contributes to spikelet fertility and thus has pleiotropic effects in shaping inflorescence architecture and facilitating floral organ development[17]. To mitigate the trade-off between spikelet number per panicle and spikelet fertility that results from the pleiotropy of *OsER1*, it is urgent

to identify specific ligands of the OsEPF/OsEPFL family that recognize OsER1 and are closely associated with panicle morphogenesis and development.

Here, we characterized the role of the *OsEPF/OsEPFL* gene family in directing rice panicle architecture and identified specific ligands of OsER1. Our results demonstrated that OsEPFL6, OsEPFL7, OsEPFL8, and OsEPFL9 synergistically contribute to rice panicle morphogenesis by recognizing OsER1 and subsequently activating the mitogen-activated protein kinase (MAPK) cascade. Interestingly, in contrast to the other three genes, *OsEPFL8* could especially control rice spikelet fertility, demonstrating the pleiotropic role of the OsEPFL8–OsER1 pair in rice panicle development. We generated higher-order null mutants for *OsEPFL6*, *OsEPFL7*, *OsEPFL8*, and *OsEPFL9*, and found that the *osepfl6 osepfl7 osepfl9* triple mutant had improved grain yield with increased spikelet number and normal spikelet fertility. This suggests that specifically suppressing the OsEPFL6–OsER1, OsEPFL7–OsER1, and OsEPFL9–OsER1 ligand–receptor pairs can optimize rice panicle architecture. These findings not only advance our understanding of how the perception of these specific SSPs by OsER1 shapes panicle morphology in rice, but also provide a framework for a fundamental understanding of the roles of ligand–receptor signaling in the developmental plasticity of plant inflorescences. Notably, our study also offers a rational molecular design to optimize crop yield traits by targeted manipulation of ligand–receptor pairs, which overcomes the trade-offs among complex traits and thus improves crop yield.

## Results

### SSPs from the *OsEPF/OsEPFL* gene family are required for rice panicle morphogenesis

The receptor-like protein kinase (RLK) OsER1 plays a negative regulatory role in determining rice spikelet number per panicle[17]. Our previous study showed that the OsER1–OsMKKK10–OsMKK4–OsMPK6 pathway controls rice panicle development by regulating cytokinin metabolism; this raises the question of which kinds of SSPs required for panicle morphogenesis can be recognized by the OsER1 receptor[17]. Cumulative evidence suggests that EPF1, EPF2, and EPFL9 from the conserved EPF/EPFL family act as ER ligands to specify stomatal development in *Arabidopsis*[19–21]. This implies that the rice OsEPF/OsEPFL family could also be recognized by OsER1 as specific SSP ligands. We therefore analyzed the rice *OsEPF/OsEPFL* gene family and found 12 homologous members: *OsEPF1*, *OsEPF2*, *OsEPFL1*, *OsEPFL2*, *OsEPFL3*, *OsEPFL4*, *OsEPFL5*, *OsEPFL6*, *OsEPFL7*, *OsEPFL8*, *OsEPFL9*, and *OsEPFL10*. These are related to the *Arabidopsis* EPF/EPFl family and encode cysteine-rich SSPs (Supplementary Fig. 1a). Another cysteine-rich SSP homolog, RAE2/GAD1, has been reported to regulate awn development during rice domestication; however, most cultivated rice has loss-of-function mutations in *RAE2/GAD1*[14,15]. We then assayed the relative expression of *OsEPF/OsEPFL* family members in the young panicles of wild-type cultivated Fengaizhan-1 (FAZ1) rice (*Oryza indica*). *OsEPF1*, *OsEPF2*, *OsEPFL1*, *OsEPFL2*, *OsEPFL3*, and *OsEPFL4* were minimally expressed during young panicle development, whereas *OsEPFL5*, *OsEPFL6*, *OsEPFL8*, and *OsEPFL9* were highly expressed (Supplementary Fig. 2a). We also examined the expression patterns of *OsEPFL6*, *OsEPFL7*, *OsEPFL8*, and *OsEPFL9* in transgenic plants expressing a GUS fusion protein driven by the promoter of each gene, and found that *OsEPFL6*, *OsEPFL8*, and *OsEPFL9* were widely and highly expressed in various organs with a pattern similar to *OsER1*. *OsEPFL7* was expressed at relatively low levels, especially in the young panicle (Supplementary Fig. 2b–f). Taken together, these results imply that *OsEPFL5* and the *OsEPFL6/7/8/9* subfamily may be the most dominant members of the *OsEPF/OsEPFL* family and may contribute to rice panicle morphogenesis.

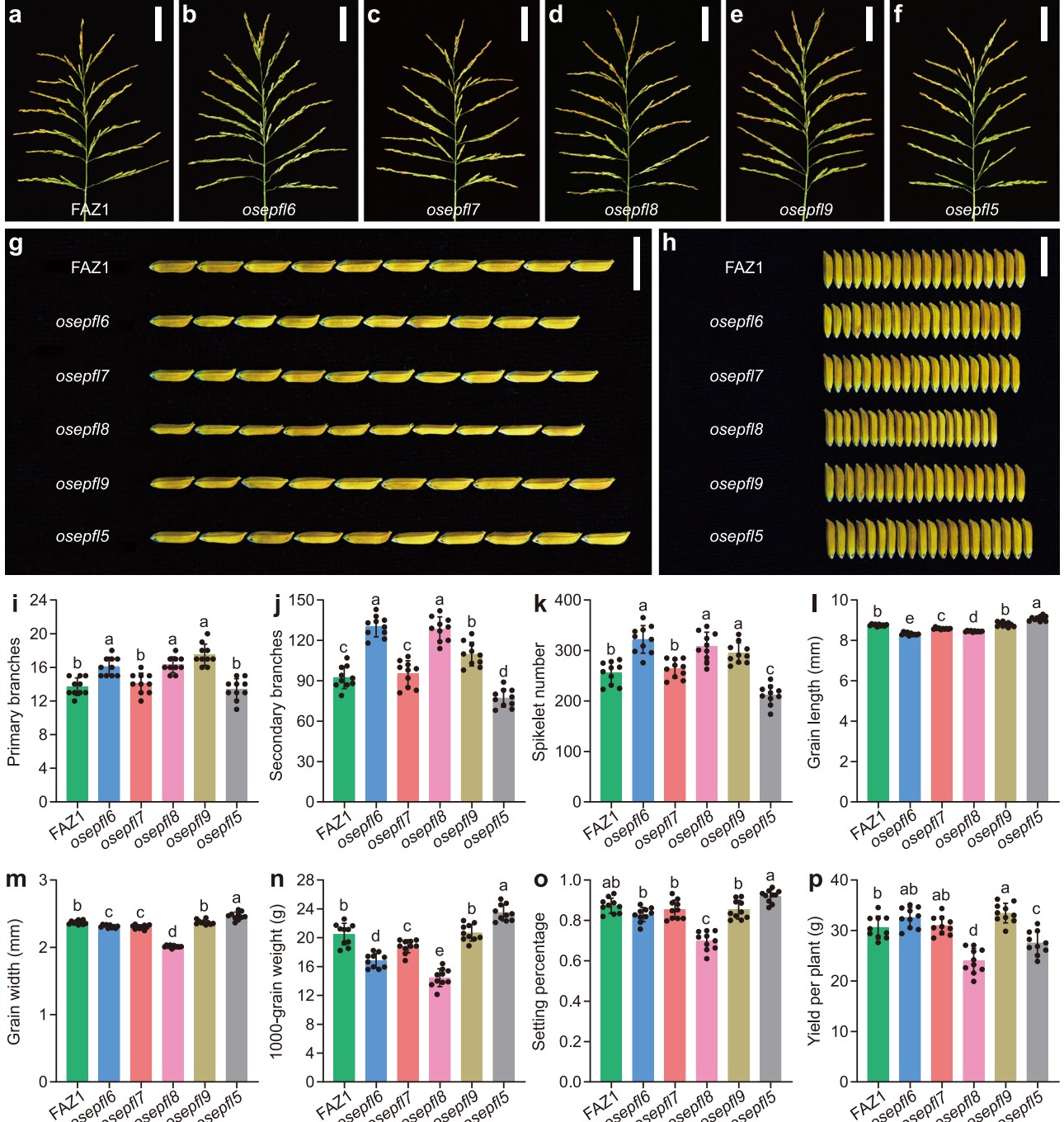

**Fig. 1 | *OsEPFL6, OsEPFL7, OsEPFL8, OsEPFL9,* and *OsEPFL5* are responsible for rice panicle morphogenesis. a–f** Rice panicles from FAZ1 (**a**) and the *osepfl6* (**b**), *osepfl7* (**c**), *osepfl8* (**d**), *osepfl9* (**e**), and *osepfl5* (**f**) mutants. Scale bar = 5 cm. **g** Comparison of rice grain length between FAZ1 and *osepfl6, osepfl7, osepfl8, osepfl9,* and *osepfl5* mutants. Scale bar = 1 cm. **h** Comparison of rice grain width between FAZ1 and *osepfl6, osepfl7, osepfl8, osepfl9,* and *osepfl5* mutants. Scale bar = 1 cm. **i–p** Comparison of the average number of primary branches (**i**), number

of secondary branches (**j**), spikelet number per panicle (**k**), grain length (**l**), grain width (**m**), 1000-grain weight (**n**), setting percentage (**o**), and yield per plant (**p**) between FAZ1 and *osepfl6, osepfl7, osepfl8, osepfl9,* and *osepfl5* mutants. Values are given as the mean ± standard deviation (SD) ($n = 10$ plants). Different letters indicate statistical significance groups at $p < 0.05$ (one-way ANOVA with post hoc Tukey's multiple comparison test). The source data underlying the statistical analysis in **i–p** are provided in the Source Data file.

To further investigate the function of SSPs in the *OsEPF/OsEPFL* family, we generated single loss-of-function mutants for *OsEPFL6, OsEPFL7, OsEPFL8, OsEPFL9,* and *OsEPFL5* using CRISPR/Cas9 gene editing[27]. The *osepfl6, osepfl7, osepfl8, osepfl9,* and *osepfl5* null mutants displayed altered panicle architecture or grain size (Fig. 1a–h and Supplementary Fig. 3, Fig. 4a–f, Fig. 5). Specifically, the *osepfl6* mutant showed increased spikelet number per panicle but reduced grain size and plant height (Fig. 1a, b, g, h and Supplementary Figs. 4a, b, and 6a,

b). Moreover, the average number of primary and secondary branches, and spikelet number per panicle of *osepfl6* mutants were markedly increased, whereas grain length and width, 1000-grain weight, and plant height were decreased compared with wild-type FAZ1 (referred to hereafter as FAZ1) (Fig. 1i–n and Supplementary Fig. 4g–l, Fig. 6g). The unaffected spikelet fertility of the *osepfl6* mutant meant that the average grain yield per plant was comparable to that of FAZ1 (Fig. 1o, p). *OsEPFL7* shares the highest sequence identity with *OsEPFL6*, and the

*osepfl7* mutant did not have a significantly enhanced spikelet number per panicle (Fig. 1a, c, i–k and Supplementary Fig. 1a, b). Both the grain weight and plant height of *osepfl7* mutants were somewhat reduced (Fig. 1g, h, l–n and Supplementary Fig. 6a, c, g), but the setting percentage and grain yield per plant of *osepfl7* were unaffected (Fig. 1o, p). These results suggest that *OsEPFL6* controls both the spikelet number and grain size, and that *OsEPFL7* primarily contributes to grain size.

Notably, although *osepfl8* mutants displayed enlarged panicles and increased spikelet number per panicle (as seen in the *osepfl6* mutants), the grain size and plant height were dramatically decreased (Fig. 1a, d, g, h and Supplementary Fig. 6a, d). Accordingly, *osepfl8* mutants had significantly increased average number of primary and secondary branches, and spikelet number per panicle, but decreased grain length and width, 1000-grain weight, and plant height compared with FAZ1 (Fig. 1i–n and Supplementary Fig. 6g). Surprisingly, *osepfl8* mutants had markedly reduced grain yield per plant due to the reduced setting percentage (Fig. 1o, p). This indicated that *OsEPFL8* specially controls rice spikelet fertility. In contrast to *osepfl8* mutants, *osepfl9* mutants had enhanced spikelet number per panicle but unaltered grain size and plant architecture (Fig. 1a, e, g, h and Supplementary Fig. 6a, e). Furthermore, the average number of primary and secondary branches, and spikelet number per panicle of *osepfl9* mutants were distinctly increased without decreased grain length or width, 1000-grain weight, or plant height compared with FAZ1 (Fig. 1i–n and Supplementary Fig. 6g). Due to the unaffected spikelet fertility, the average grain yield per plant of *osepfl9* mutants was significantly improved (Fig. 1o, p), suggesting that *OsEPFL9* specifically contributes to spikelet number.

*OsEPFL5* was highly expressed in the young panicle, similar to members of the *OsEPFL6/7/8/9* subfamily (Supplementary Fig. 2a). However, loss of *OsEPFL5* function resulted in different panicle phenotypes compared to the *osepfl6/7/8/9* mutants, including reduced spikelet number per panicle and increased grain size (Fig. 1a, f, g, h). The average number of primary branches was comparable to those of FAZ1, but the average number of secondary branches were significantly decreased, and the grain length and width and 1000-grain weight were higher than in FAZ1 (Fig. 1i–n). Additionally, *osepfl5* mutants had reduced plant height compared with FAZ1 (Supplementary Fig. 6a, f, g). Although the setting percentage was unchanged, *osepfl5* mutants had lower average grain yield per plant (Fig. 1o, p). These results suggested that *OsEPFL5* has a role that is contrary to those of the *OsEPFL6/7/8/9* subfamily members in controlling spikelet number and grain size. Consistent with this finding, the predicted mature OsEPFL5 peptide was largely different from those of OsEPFL6, OsEPFL7, OsEPFL8, and OsEPFL9, even though they are all from the cysteine-rich OsEPF/OsEPFL family (Supplementary Fig. 1b, c). This implies that OsEPFL5 has structurally differentiated from OsEPFL6, OsEPFL7, OsEPFL8, and OsEPFL9, which could alter the binding capacity and specificity of the receptor. Overall, these results suggest that *OsEPFL6*, *OsEPFL7*, *OsEPFL8*, *OsEPFL9*, and *OsEPFL5* are required for rice panicle morphogenesis, and that the *OsEPFL6/7/8/9* subfamily members are negative regulators of spikelet number per panicle, whereas *OsEPFL5* is a positive regulator.

### OsEPFL6, OsEPFL7, OsEPFL8, and OsEPFL9 synergistically contribute to rice panicle morphogenesis by activating the MAPK cascade

Our results demonstrated that the *OsEPFL6/7/8/9* subfamily and *OsEPFL5* are responsible for rice panicle morphogenesis and have distinct roles in this process. To further investigate their genetic relationships, we generated a collection of double knockout mutants: *osepfl6 osepfl7* (*osepfl6;7*), *osepfl6 osepfl8* (*osepfl6;8*), *osepfl6 osepfl9* (*osepfl6;9*), *osepfl7 osepfl8* (*osepfl7;8*), *osepfl7 osepfl9* (*osepfl7;9*), *osepfl8 osepfl9* (*osepfl8;9*), *osepfl5 osepfl6* (*osepfl5;6*), and *osepfl5 osepfl8* (*osepfl5;8*) (Fig. 2a–i). We found that panicles of *osepfl6;7* double

mutants were enlarged, similar to those of *osepfl6* but not *osepfl7* single mutants (Fig. 2a, b, m). The average number of primary and secondary branches were also increased, but both the grain length and width were reduced compared with FAZ1 (Supplementary Fig. 7a–d), and plant height was decreased compared to the *osepfl6* mutant (Supplementary Fig. 8a, b, m). Because the setting percentage was unaffected, the average grain yield per plant was comparable to those of FAZ1 and the *osepfl6* and *osepfl7* single mutants (Fig. 2n and Supplementary Fig. 7e). These results indicated that *OsEPFL6* is epistatic to *OsEPFL7* and has a key role in shaping the rice panicle. Strikingly, the *osepfl6;8* double mutant displayed more spikelets per panicle than the *osepfl6* or *osepfl8* single mutants (Fig. 2a, c, m). Although the average number of primary and secondary branches were clearly increased, the grain length and width were decreased and plant height was reduced compared with FAZ1 and the corresponding single mutants (Supplementary Fig. 7a–d, Fig. 8a, c, m). Notably, the average grain yield per plant of *osepfl6;8* double mutants was largely decreased because of the lower setting percentage (Fig. 2n and Supplementary Fig. 7e). These results suggest that *OsEPFL6* and *OsEPFL8* redundantly contribute to spikelet number per panicle and grain size, and are therefore both essential regulators of rice panicle morphogenesis. Furthermore, we found that the *osepfl6;9* double mutant exhibited increased spikelet number per panicle compared with FAZ1 and the *osepfl6* and *osepfl9* single mutants (Fig. 2a, d, m), likely due to the additional primary and secondary branches (Supplementary Fig. 7a, b). The average grain length and width and plant height of the *osepfl6;9* double mutants were comparable to those of *osepfl6* mutants, indicating that *OsEPFL6* is epistatic to *OsEPFL9* in regulating grain size and plant architecture (Supplementary Fig. 7c, d, Fig. 8a, d, m). The *osepfl6;9* double mutants had unaltered setting percentages, and the average grain yield per plant was therefore significantly enhanced compared to FAZ1, comparable to *osepfl9* mutants (Fig. 2n and Supplementary Fig. 7e). These results indicate that *OsEPFL6* and *OsEPFL9* redundantly control spikelet number per panicle, thus providing a potential means to breed high-yield rice.

The *osepfl7;8* double mutants also displayed enlarged panicles, similar to the *osepfl8* rather than the *osepfl7* single mutant (Fig. 2a, e, m). Accordingly, the average number of primary and secondary branches were increased, but both the grain length and width were reduced compared with FAZ1 (Supplementary Fig. 7a–d). The *osepfl7;8* double mutant plant height was comparable to *osepfl8* (Supplementary Fig. 8a, e, m). The *osepfl7;8* double mutants showed decreased setting percentages compared to FAZ1 and were comparable to the *osepfl8* mutants, which resulted in attenuated grain yield per plant (Fig. 2n and Supplementary Fig. 7e). These results suggest that *OsEPFL8* is epistatic to *OsEPFL7* and plays a dominant role in controlling spikelet number and grain size. Moreover, *osepfl7;9* double mutant panicles showed increased spikelet number per panicle, comparable to *osepfl9* single mutants (Fig. 2a, f, m). The average number of primary and secondary branches also increased, and grain length and width were reduced compared to FAZ1 and the *osepfl7* and *osepfl9* single mutants (Supplementary Fig. 7a–d). The *osepfl7;9* plants had reduced height compared to the *osepfl9* mutant (Supplementary Fig. 8a, f, m). Notably, the average grain yield per plant of *osepfl7;9* double mutants was significantly enhanced because the setting percentage was unaffected (Fig. 2n and Supplementary Fig. 7e). These results imply that *OsEPFL9* is also epistatic to *OsEPFL7* in determining spikelet number. The *osepfl8;9* double mutants had more spikelets per panicle than either *osepfl8* or *osepfl9* plants (Fig. 2a, g, m). The average number of primary and secondary branches were significantly increased, but the grain length and width were dramatically reduced compared with FAZ1 or the corresponding single mutants (Supplementary Fig. 7a–d). The *osepfl8;9* plants were similar in height to *osepfl8* single mutants (Supplementary Fig. 8a, g, m). Notably, the setting percentage affected the average grain yield per plant of *osepfl8;9* mutants, which was decreased

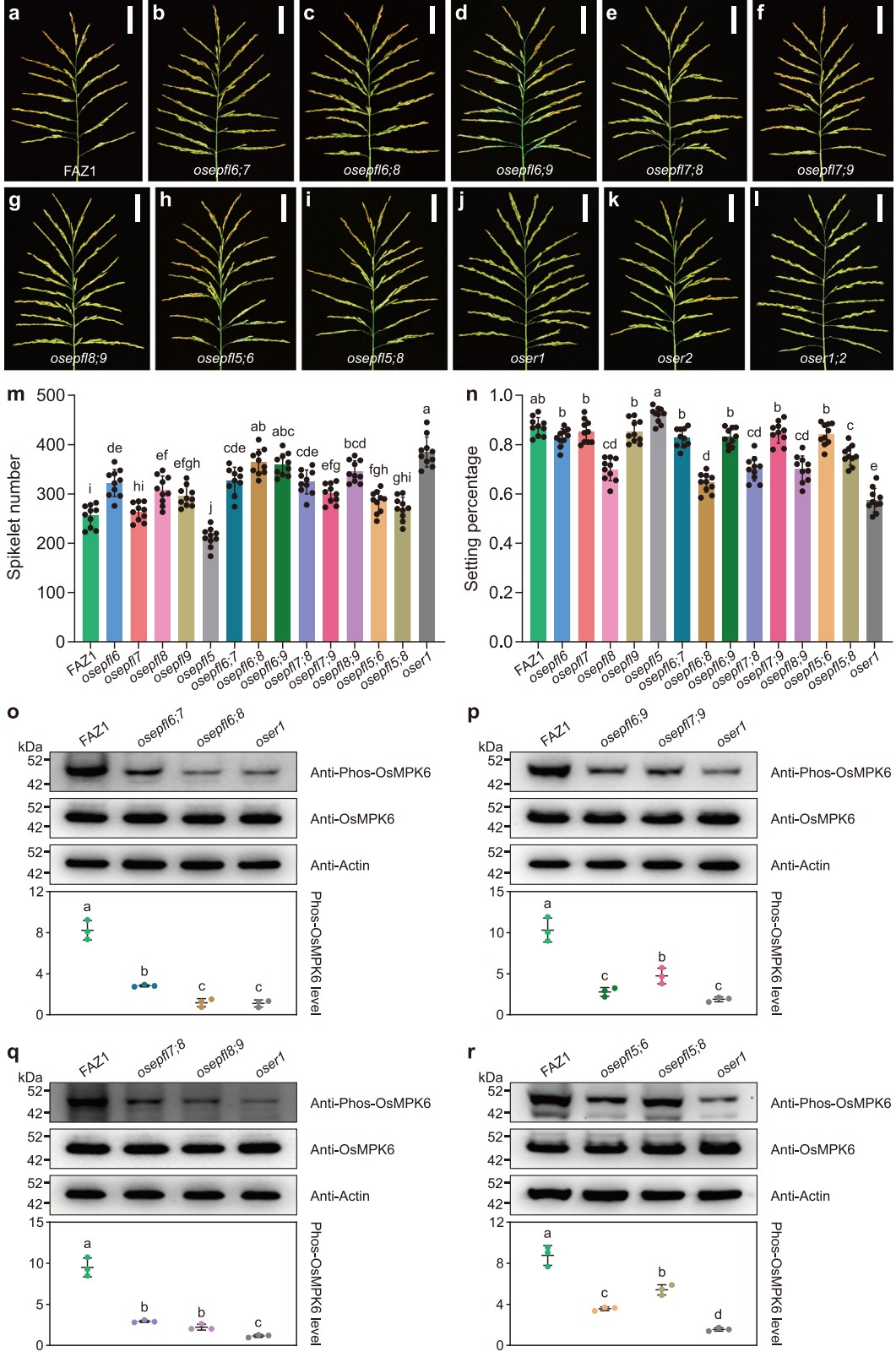

compared to *osepfl9* single mutants (Fig. 2n and Supplementary Fig. 7e). These results indicated that *OsEPFL8* and *OsEPFL9* are both pivotal regulators in determining panicle architecture, and they redundantly affect rice spikelet number per panicle. However, *OsEPFL8* is epistatic to *OsEPFL9* in modulating grain size and spikelet fertility. Overall, these findings suggest that *OsEPFL6*, *OsEPFL7*, *OsEPFL8*, and *OsEPFL9* contribute to rice panicle morphogenesis redundantly and synergistically, but also have distinct roles.

Based on the results showing the positive role of *OsEPFL5* in regulating spikelet number per panicle and the key roles of *OsEPFL6* and *OsEPFL8*, we further analyzed the *osepfl5;6* and *osepfl5;8* double mutants. As expected, the spikelet number per panicle was greatly reduced in the *osepfl5;6* double mutant compared with the *osepfl6* (Fig. 2a, h, m) and in the *osepfl5;8* double mutant compared with *osepfl8* (Fig. 2a, i, m). Consistent with these results, the average number of primary and secondary branches from the double mutants were

**Fig. 2 | *OsEPFL6, OsEPFL7, OsEPFL8,* and *OsEPFL9* redundantly control rice spikelet number per panicle by activating the MAPK cascade. a–l** Rice panicles from FAZ1 (**a**) and the *osepfl6;7* (**b**), *osepfl6;8* (**c**), *osepfl6;9* (**d**), *osepfl7;8* (**e**), *osepfl7;9* (**f**), *osepfl8;9* (**g**), *osepfl5;6* (**h**), *osepfl5;8* (**i**), *oser1* (**j**), *oser2* (**k**), and *oser1;2* (**l**) mutants. Scale bar = 5 cm. **m, n** Comparison of average spikelet number per panicle (**m**), and setting percentage (**n**) between FAZ1, *osepfl6, osepfl7, osepfl8, osepfl9, osepfl5, osepfl6;7, osepfl6;8, osepfl6;9, osepfl7;8, osepfl7;9, osepfl8;9, osepfl5;6, osepfl5;8,* and *oser1* mutants. Values are given as the mean ± SD (*n* = 10 plants). Different letters indicate statistical significance groups at *p* < 0.05 (one-way ANOVA with post-hoc Tukey's multiple comparison test). **o–r** Comparison of OsMPK6 phosphorylation levels in FAZ1, *osepfl6;7, osepfl6;8,* and *oser1* (**o**); FAZ1, *osepfl6;9, osepfl7;9,* and *oser1* (**p**); FAZ1, *osepfl7;8, osepfl8;9,* and *oser1* (**q**); and FAZ1, *osepfl5;6, osepfl5;8,* and *oser1* (**r**). Proteins extracted from young panicles were analyzed with immunoblot using anti-Phos-OsMPK6 and anti-OsMPK6 antibodies. Anti-Actin antibody was used as the loading control. The graphs show the qualification of the relative levels of the phosphorylated OsMPK6. Values are given as the mean ± SD (*n* = 3 biological replicates). Different letters indicate statistical significance groups at *p* < 0.05 (one-way ANOVA with post-hoc Tukey's multiple comparison test). The source data underlying the statistical analysis in **m–r** and uncropped images in **o–r** are provided in the Source Data file.

increased compared to the *osepfl5* single mutant (Supplementary Fig. 7a, b). However, both *osepfl5;6* and *osepfl5;8* had significantly reduced grain length and width compared to the *osepfl5* mutant (Supplementary Fig. 7c, d). Plant height was comparable between the double and the corresponding single mutants (Supplementary Fig. 8a, h, i, m). Although the setting percentage was decreased in both double mutants compared with *osepfl5*, the grain yield per plant was comparable (Fig. 2n and Supplementary Fig. 7e). These results confirmed that *OsEPFL5* plays an opposing role in rice panicle morphogenesis compared with *OsEPFL6* and *OsEPFL8*. Thus, their functions could be mutually exclusive and may determine by the corresponding receptors.

Notably, the phenotypes of the double mutants were reminiscent of those of the *oser1* null mutant[17]. We thus hypothesized that OsER1 may act as a potential receptor recognized by mature OsEPFL6, OsEPFL7, OsEPFL8, and OsEPFL9 small peptide ligands. We further compared the spikelet number per panicle between the double mutants and the *oser1* null mutant, and found that only the *osepfl6;8* and *osepfl6;9* double mutants were similar to *oser1* (Fig. 2a–j, m). Accordingly, the *osepfl6;8* but not the *osepfl6;9* double mutant had fewer primary branches than *oser1*; however, both double mutants had a similar number of secondary branches to the *oser1* mutant (Supplementary Fig. 7a, b). These results suggested that *OsEPFL6, OsEPFL8,* and *OsEPFL9* dominate their subfamily with specific functions. Comprehensive comparisons of the setting percentage, grain size, grain yield per plant, and plant height between the double mutants and the *oser1* null mutant supported that the *OsEPFL6/7/8/9* subfamily, but not *OsEPFL5*, was positively associated with *OsER1* (Fig. 2n and Supplementary Fig. 7c–e, Fig. 8m). Nevertheless, due to the possible redundancy between *OsER1* and its homolog *OsER2*, we generated an *oser2* single mutant and an *oser1;2* double mutant. The *oser2* mutant had panicles comparable to FAZ1, but *oser1;2* panicles were similar to those of *oser1* (Fig. 2a, j, k, l and Supplementary Fig. 9a). Although the number of primary and secondary branches of *oser2* were somewhat more than those of FAZ1, the spikelet number per panicle, grain length, and width, setting percentage, and yield per plant were comparable (Supplementary Fig. 9b–i). Notably, yield traits of *oser1* plants were comparable to those of *oser1;2* (Fig. 2a, j, k, l and Supplementary Fig. 9b–i), indicating that *OsER1* was epistatic to *OsER2* and dominated rice panicle development. Taken together, these results indicate that the *OsEPFL6/7/8/9* subfamily redundantly shapes panicle architecture resembling the function of *OsER1* rather than *OsER2*. This implies that mature OsEPFL6, OsEPFL7, OsEPFL8, and OsEPFL9 could act as small peptide ligands of the OsER1 receptor.

In a previous study, we demonstrated that loss of *OsER1* function suppresses phosphorylation of OsMPK6 and that OsER1 acts upstream of the OsMKKK10–OsMKK4–OsMPK6 cascade to negatively regulate spikelet number per panicle[17]. Therefore, we here further assayed the level of OsMPK6 phosphorylation in young panicles of single and double mutants. Levels of OsMPK6 phosphorylation were clearly attenuated in *osepfl6, osepfl8,* and *osepfl9* mutants. Unexpectedly, OsMPK6 phosphorylation levels were higher in *osepfl5* mutants compared with FAZ1 (Supplementary Fig. 10), implying that the loss of *OsEPFL5* function could activate the MAPK cascade. Notably, levels of

OsMPK6 phosphorylation were reduced in the young panicles of *osepfl6;7* and *osepfl6;8* double mutants despite their comparable protein abundance with FAZ1; OsMPK6 phosphorylation levels in the *osepfl6;8* mutant were similar to those of the *oser1* null mutant (Fig. 2o). Similarly, OsMPK6 phosphorylation was decreased in *osepfl6;9* and *osepfl7;9* double mutants (Fig. 2p). As expected, the *osepfl7;8* and *osepfl8;9* double mutants also exhibited attenuated OsMPK6 phosphorylation levels, comparable to those of the *oser1* single mutant (Fig. 2q). These results indicated that members of the *OsEPFL6/7/8/9* subfamily are required for phosphorylation of OsMPK6 and could trigger the OsMKKK10–OsMKK4–OsMPK6 cascade. Because OsMPK6 phosphorylation levels are closely associated with spikelet number per panicle and grain size[3], we also assayed OsMPK6 phosphorylation in the *osepfl5;6* and *osepfl5;8* double mutants. This showed that OsMPK6 phosphorylation was somewhat decreased in these mutants compared with FAZ1, particularly in the *osepfl5;6* double mutant (Fig. 2r). This was consistent with the morphological phenotypes of these double mutants and suggests a negative role of *OsEPFL5* in the OsER1–MAPK pathway. These results imply that OsMPK6 phosphorylation and activation of OsER1–OsMKKK10–OsMKK4–OsMPK6 signaling may be independent of *OsEPFL5*, or that there is a negative feedback mechanism for finely controlling levels of OsMPK6 phosphorylation for rice panicle development. In summary, these results suggest that members of the *OsEPFL6/7/8/9* subfamily redundantly and synergistically contribute to rice panicle morphogenesis by activating the MAPK cascade in a manner similar to that of *OsER1*.

## OsEPFL6, OsEPFL7, OsEPFL8, and OsEPFL9 act as ligands of the OsER1 receptor

Although we hypothesized that mature OsEPFL6, OsEPFL7, OsEPFL8, and OsEPFL9 proteins could act as small peptide ligands to recognize the OsER1 receptor, their exact molecular functions have remained largely unknown. To address this question, we in vitro expressed and purified the extracellular leucine-rich repeat (LRR) domain of OsER1 (OsER1$^{LRR}$), and the predicted mature small peptides OsEPFL6, OsEPFL7, OsEPFL8, and OsEPFL9 from insect cells (Supplementary Fig. 1b). We then investigated whether the small peptide ligands were capable of interacting with OsER1$^{LRR}$ using isothermal titration calorimetry (ITC) technique. OsEPFL6 and OsEPFL7 strongly interacted with OsER1$^{LRR}$, with dissociation constants of 1.35 μM and 1.02 μM, respectively (Fig. 3a, b). OsEPFL8 and OsEPFL9 also interacted with OsER1$^{LRR}$, with dissociation constants of 18.73 μM and 56.82 μM, respectively (Fig. 3c, d), indicating that mature OsEPFL8 and OsEPFL9 had different receptor affinities from OsEPFL6 and OsEPFL7. These results suggest that the small peptides OsEPFL6, OsEPFL7, OsEPFL8, and OsEPFL9 could bind to OsER1$^{LRR}$ in vitro. The ITC results were further supported by gel filtration assays, in which OsER1$^{LRR}$ clearly interacted with OsEPFL6, OsEPFL7, OsEPFL8, and OsEPFL9, as indicated by their co-migration (Fig. 3e). Taken together, our results indicate that the small peptides OsEPFL6, OsEPFL7, OsEPFL8, and OsEPFL9 could act as ligands to recognize the extracellular domain of the OsER1 receptor in vitro.

To further verify the ITC and gel filtration assay results, we performed co-immunoprecipitation (Co-IP) experiments with OsER1 and

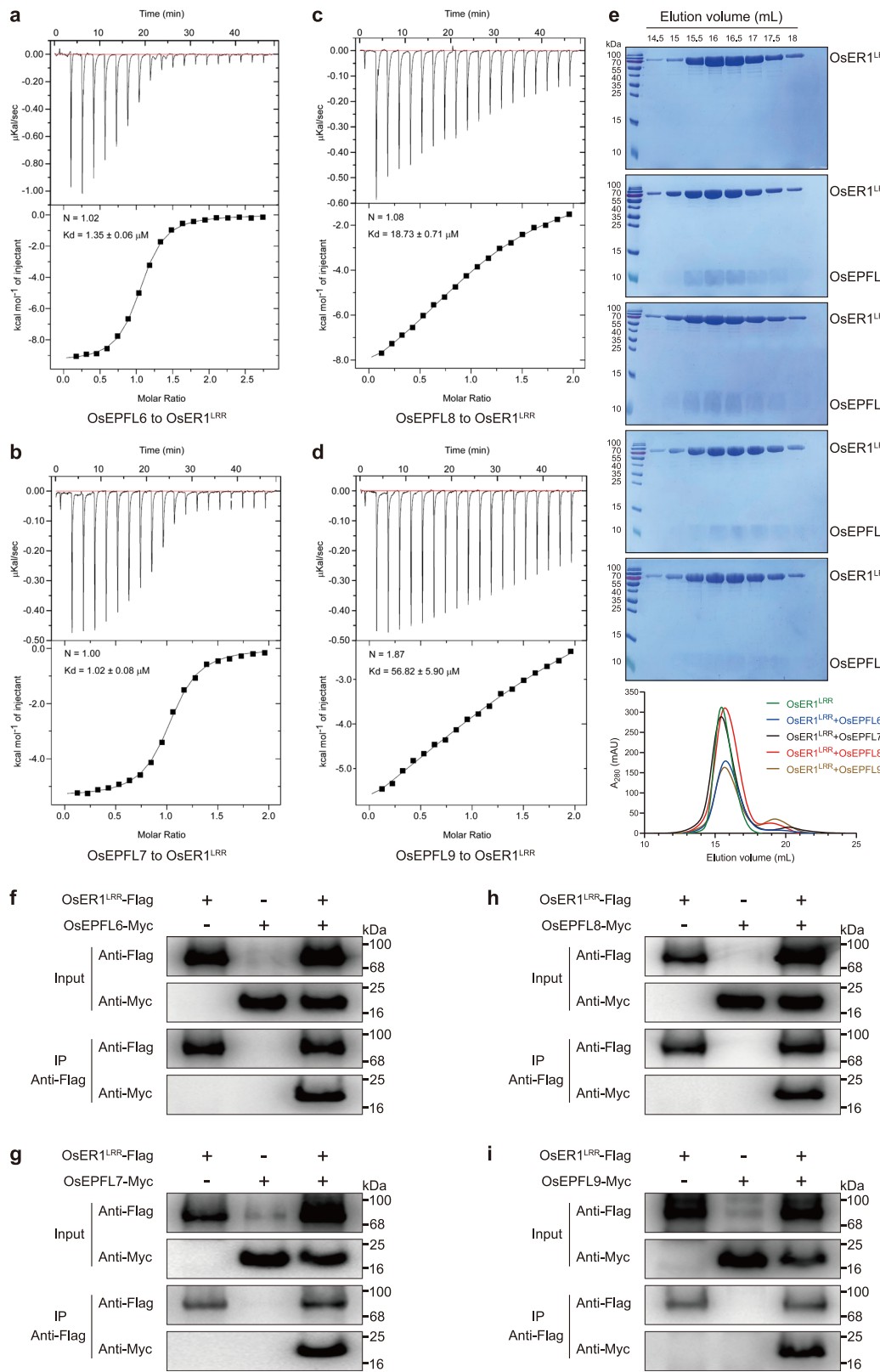

OsEPFL6, OsEPFL7, OsEPFL8, and OsEPFL9. Full-length OsER1 was relatively unstable[19]. Thus, OsER1$^{LRR}$–Flag was co-expressed in *Nicotiana benthamiana* leaves with either OsEPFL6-Myc, OsEPFL7-Myc, OsEPFL8–Myc, or OsEPFL9–Myc fusion protein. Protein extracts from the tobacco leaves were then immunoprecipitated with anti-Flag beads and detected in an immunoblot analysis with anti-Flag and anti-Myc antibodies. Notably, we found that a band with the expected size of

OsEPFL6–Myc was detected in the anti-Flag immunoprecipitate from leaves expressing OsER1$^{LRR}$–Flag and OsEPFL6-Myc; in contrast, OsEPFL6-Myc was not detected in the absence of OsER1$^{LRR}$–Flag expression (Fig. 3f). As expected, OsEPFL7-Myc, OsEPFL8–Myc, and OsEPFL9–Myc were also detected in the anti-Flag immunoprecipitate from leaves co-expressing OsER1$^{LRR}$–Flag (Fig. 3g–i). This indicated that the small peptides OsEPFL6, OsEPFL7, OsEPFL8, and OsEPFL9

**Fig. 3 | OsEPFL6, OsEPFL7, OsEPFL8, and OsEPFL9 act as ligands to directly bind the OsER1 receptor. a–d** Quantification of the binding affinity of OsER1^LRR for OsEPFL6 (**a**), OsEPFL7 (**b**), OsEPFL8 (**c**), and OsEPFL9 (**d**) as measured with iso-thermal titration calorimetry (ITC) assays. Purified OsEPFL6, OsEPFL7, OsEPFL8, and OsEPFL9 small peptides were separately titrated into OsER1^LRR protein in the ITC cell. Raw data and integrated heat measurements are shown in the upper and lower panels, respectively. The calculated stoichiometry (N) and the dissociation constants (Kd) are indicated. **e** Gel filtration showing that OsEPFL6, OsEPFL7, OsEPFL8, and OsEPFL9 directly bind to OsER1^LRR. The elution volumes and the molecular weight markers are indicated at the top. Size exclusion chromatography analysis of the interactions between OsEPFL6 and OsER1^LRR, OsEPFL7 and OsER1^LRR, OsEPFL8, and OsER1^LRR, and OsEPFL9 and OsER1^LRR are shown in the lower panel.

The middle panel shows peak fractions from the lower panel analyzed by SDS-PAGE with Coomassie brilliant blue staining corresponding to OsER1^LRR alone, OsEPFL6–OsER1^LRR, OsEPFL7–OsER1^LRR, OsEPFL8–OsER1^LRR, and OsEPFL9–OsER1^LRR. **f–i** Co-IP assays indicating that OsEPFL6 (**f**), OsEPFL7 (**g**), OsEPFL8 (**h**), and OsEPFL9 (**i**) each interact with OsER1^LRR in planta. Pro35S::OsER1^LRR-Flag was co-expressed with Pro35S::OsEPFL6-Myc, Pro35S::OsEPFL7-Myc, Pro35S::OsEPFL8-Myc, or Pro35S::OsEPFL9-Myc in *N. benthamiana* leaves. Proteins were extracted (Input) and immunoprecipitated (IP) with Flag beads. Immunoblots were performed using anti-Flag and anti-Myc antibodies. In **e–i**, three independent experiments were repeated with similar results. The source data underlying the statistical analysis in **e** and uncropped images in **f–i** are provided in the Source Data file.

interacted with OsER1 in vivo. Overall, these results demonstrate that OsEPFL6, OsEPFL7, OsEPFL8, and OsEPFL9 act as ligands to recognize the OsER1 receptor both in vitro and in vivo.

We next confirmed the physiological activity of the purified small peptides. The mature OsEPFL6 small peptide was chosen as a representative to determine whether it could promote rice seedling growth. We found that different concentrations of the mature OsEPFL6 small peptide could promote elongation of the shoot but not the root of FAZ1 seedlings; however, it failed to promote shoot elongation of the *oser1* mutant (Supplementary Fig. 11). This indicated not only that the purified mature OsEPFL6 small peptide has bioactivity, but also that its physiological function is dependent on *OsER1*, thereby suggesting that the OsEPFL6 small peptide facilitated rice growth and development and is dependent on the OsER1 signaling pathway.

### OsEPFL6, OsEPFL7, OsEPFL8, and OsEPFL9 negatively regulate spikelet number per panicle in an OsER1-dependent manner

Our findings provide evidence that the OsEPFL6, OsEPFL7, OsEPFL8, and OsEPFL9 small peptides act as ligands to recognize the OsER1 receptor. However, the genetic relationship between the *OsEPFL6/7/8/9* subfamily and *OsER1* remained unknown. We thus generated the *oser1 osepfl6 osepfl8* (*oser1 osepfl6;8*) triple mutant, which exhibited enlarged panicle architecture comparable to the *oser1* and *osepfl6;8* mutants (Fig. 4a–d). The *oser1 osepfl6;8* triple mutants had a higher average spikelet number per panicle compared to the *osepfl6;8* double mutant, comparable to that of *oser1*. This indicated that *OsER1* was epistatic to *OsEPFL6* and *OsEPFL8* (Fig. 4m). Furthermore, to determine whether *OsEPFL6*, *OsEPFL7*, *OsEPFL8*, and *OsEPFL9* act in the *OsER1* pathway, we tested whether constitutive overexpression of *OsEPFL6*, *OsEPFL7*, *OsEPFL8*, or *OsEPFL9* was *OsER1*-dependent. Overexpression of *OsEPFL6* in FAZ1 resulted in production of fewer spikelets per panicle, but had no effect on spikelet number per panicle in the *oser1* mutant background (Fig. 4a, e–h, n, o and Supplementary Fig. 12a, b). Similar phenotypes were also observed in *OsEPFL8*-overexpression lines (Fig. 4a, i–l, n, o and Supplementary Fig. 12c, d), indicating that *OsEPFL6* and *OsEPFL8* are dependent on *OsER1* to negatively control spikelet number per panicle. Although loss of *OsEPFL7* function primarily contributed to grain size (Fig. 1), we also generated *OsEPFL7*-overexpression lines in FAZ1, which showed a reduced spikelet number per panicle, similar to plant overexpressing *OsEPFL6* or *OsEPFL8* (Supplementary Fig. 13a–c, f, h). Similarly, *OsEPFL9* overexpression resulted in fewer spikelets per panicle (Supplementary Fig. 13a, d, e, g, h). These results show that *OsEPFL7* and *OsEPFL9* negatively regulate spikelet number per panicle.

Because the *OsEPFL6/7/8/9* subfamily is required for MAPK activation in addition to *OsER1*, we assayed OsMPK6 phosphorylation levels in the young panicles of the *oser1* single and higher-order mutants. OsMPK6 phosphorylation was greatly decreased in the young panicle of *oser1 osepfl6;8* mutants, with levels similar to those seen in *oser1* and *osepfl6;8* mutants (Fig. 4p). Moreover, constitutive over-expression of either *OsEPFL6* or *OsEPFL8* increased OsMPK6 phosphorylation in the FAZ1 background (Fig. 4q, r); however, they failed to

increase OsMPK6 phosphorylation in the *oser1* mutant background (Fig. 4s). These results suggest that *OsEPFL6* and *OsEPFL8* are dependent on MAPK activation to negatively regulate spikelet number per panicle. OsMPK6 phosphorylation levels were also increased in *OsEPFL7*- and *OsEPFL9*-overexpression lines compared with FAZ1 (Supplementary Fig. 13i), indicating that both *OsEPFL7* and *OsEPFL9* act as negative modulators of spikelet formation dependent on activation of the MAPK cascade. Overall, these results suggest that *OsEPFL6*, *OsEPFL7*, *OsEPFL8*, and *OsEPFL9* negatively regulate spikelet number per panicle and are genetically dependent on the *OsER1* pathway.

### Optimizing panicle architecture to improve rice yield by suppressing ligand–receptor pairs

Our results thus far demonstrated that *OsER1* plays a negative role in determining spikelet number per panicle; loss of *OsER1* function dramatically enhanced spikelet number per panicle. However, spikelet fertility was compromised in the *oser1* null mutant, indicating that *OsER1* is pleiotropic in shaping panicle architecture[17]. Interestingly, *OsEPFL8* has been found to control rice spikelet fertility (Fig. 1o). This implies that there is a trade-off between spikelet number per panicle and spikelet fertility that could be overcome to optimize rice panicle architecture by manipulating the *OsEPFL6/7/8/9* subfamily and bypassing the loss of *OsER1* function. We thus generated a collection of triple and quadruple mutants for *OsEPFL6*, *OsEPFL7*, *OsEPFL8*, and *OsEPFL9* to assess yield traits. The *osepfl6 osepfl7 osepfl8* (*osepfl6;7;8*), *osepfl6 osepfl7 osepfl9* (*osepfl6;7;9*), *osepfl6 osepfl8 osepfl9* (*osepfl6;8;9*), *osepfl7 osepfl8 osepfl9* (*osepfl7;8;9*), and *osepfl6 osepfl7 osepfl8 osepfl9* (*osepfl6;7;8;9*) mutants all showed enlarged panicle architecture and increased spikelet number per panicle, comparable to the *oser1* null mutant (Fig. 5a–h). In particular, the *osepfl6;7;8;9* quadruple mutant was almost identical to the *oser1* mutant (Fig. 5f–h), further confirming that *OsEPFL6*, *OsEPFL7*, *OsEPFL8*, and *OsEPFL9* depend on *OsER1* to control panicle morphogenesis.

The average number of primary and secondary branches were significantly increased in the *osepfl6;7;8*, *osepfl6;8;9*, and *osepfl7;8;9* triple mutants (Supplementary Fig. 14a, b), but the setting percentage and grain size were compromised, much like in the *osepfl6;7;8;9* and *oser1* mutants (Fig. 5i and Supplementary Fig. 14c, d). However, the setting percentage of the *osepfl6;7;9* triple mutant was unaffected (Fig. 5i). Only the *osepfl6;7;9* triple mutant had markedly enhanced grain yield per plant (-14.4%), and thereby elevated plot yield (-7.3%) (Fig. 5j, k). This indicated that genetic manipulation of the *OsEPFL6/7/8/9* subfamily could improve rice yield. Intriguingly, plant height was reduced in the quadruple mutant and in all four triple mutants compared with FAZ1 (Supplementary Fig. 15a–g, o); the rice stems of the triple and quadruple mutants were thicker than those of FAZ1 but comparable to those of *oser1* plants (Supplementary Fig. 15h–n, p), although the tiller number had no significant difference (Supplementary Fig. 15q). This implied that the loss of *OsEPFL6*, *OsEPFL7*, *OsEPFL8*, or *OsEPFL9* function could optimize rice plant architecture. We also found that OsMPK6 phosphorylation levels were decreased in the young panicles of triple and quadruple mutants, confirming not only

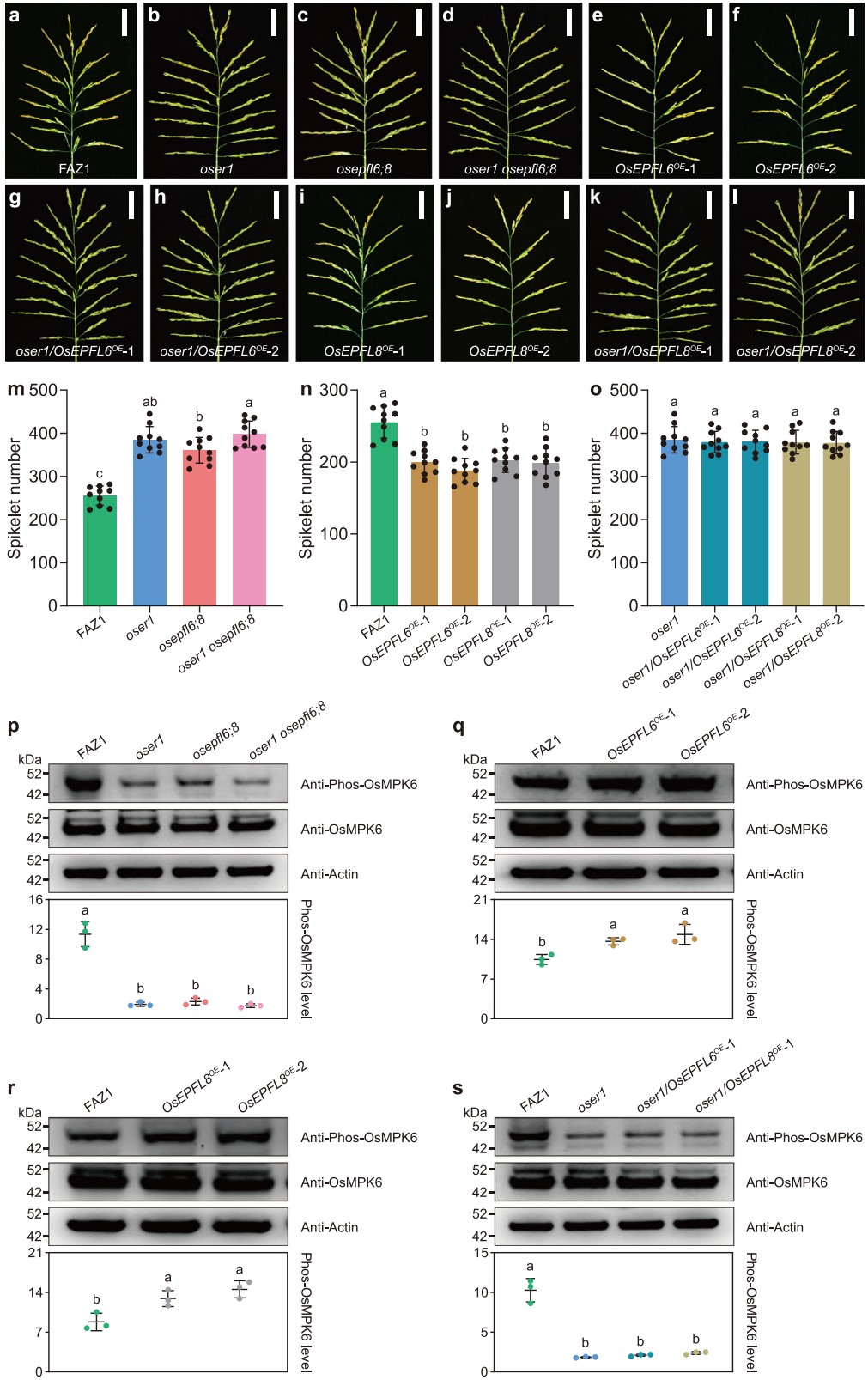

that *OsEPFL6*, *OsEPFL7*, *OsEPFL8*, and *OsEPFL9* synergistically contribute to rice panicle morphogenesis by activating the MAPK cascade, but also suggesting that attenuation of OsMPK6 phosphorylation levels could be beneficial for high-yield panicle architecture (Supplementary Fig. 16). Taken together, these results suggest that panicle architecture can be optimized to enhance rice grain yield by suppressing the OsEPFL6–OsER1, OsEPFL7–OsER1, and OsEPFL9–OsER1

ligand–receptor pairs, thus overcoming the trade-offs between yield traits resulting from the loss of *OsER1* function.

## Discussion

Rice is a model plant in the grass family. It has evolved a characteristic panicle architecture with hierarchical primary and secondary branches and specialized spikelets, which are closely related

**Fig. 4 | *OsEPFL6* and *OsEPFL8* negatively regulate spikelet number per panicle and are genetically dependent on the *OsER1* pathway. a–d** Rice panicles from FAZ1 (**a**) and the *oser1* (**b**), *osepfl6;8* (**c**), and *oser1 osepfl6;8* (**d**) mutants. **e–f** Panicles from the *OsEPFL6*-overexpression lines *OsEPFL6$^{OE}$*-1 (**e**) and *OsEPFL6$^{OE}$*-2 (**f**) in the FAZ1 background. **g–h** Panicles from the *OsEPFL6*-overexpression lines *oser1/OsEPFL6$^{OE}$*-1 (**g**) and *oser1/OsEPFL6$^{OE}$*-2 (**h**) in the *oser1* mutant background. **i–j** Panicles from *OsEPFL8*-overexpression lines *OsEPFL8$^{OE}$*-1 (**i**) and *OsEPFL8$^{OE}$*-2 (**j**) in the FAZ1 background. **k–l** Panicles from *OsEPFL8*-overexpression lines *oser1/OsEPFL8$^{OE}$*-1(**k**) and *oser1/OsEPFL8$^{OE}$*-2 (**l**) in the *oser1* mutant background. Scale bar = 5 cm. **m–o** Comparison of average spikelet number per panicle between FAZ1, *oser1*, *osepfl6;8*, and *oser1 osepfl6;8* (**m**); FAZ1, *OsEPFL6$^{OE}$*–1, *OsEPFL6$^{OE}$*–2, *OsEPFL8$^{OE}$*–1, and *OsEPFL8$^{OE}$*–2 (**n**); and FAZ1, *oser1/OsEPFL6$^{OE}$*–1, *oser1/OsEPFL6$^{OE}$*–2, *oser1/OsEPFL8$^{OE}$*–1, and *oser1/OsEPFL8$^{OE}$*–2 (**o**). Values are shown as the mean ± SD (*n* = 10

plants). Different letters indicate statistical significance groups at *p* < 0.05 (one-way ANOVA with post-hoc Tukey's multiple comparison test). **p–s** Comparison of OsMPK6 phosphorylation levels in FAZ1, *oser1*, *osepfl6;8*, and *oser1 osepfl6;8* (**p**); FAZ1, *OsEPFL6$^{OE}$*–1, *OsEPFL6$^{OE}$*–2 (**q**); FAZ1, *OsEPFL8$^{OE}$*–1, *OsEPFL8$^{OE}$*–2 (**r**); and FAZ1, *oser1*, *oser1/OsEPFL6$^{OE}$*–1, and *oser1/OsEPFL8$^{OE}$*–1 (**s**). Proteins extracted from young panicles were analyzed via immunoblot using anti-Phos-OsMPK6 and anti-OsMPK6 antibodies. Anti-Actin antibody was used as the loading control. The graphs show the qualification of the relative levels of the phosphorylated OsMPK6. Values are given as the mean ± SD (*n* = 3 biological replicates). Different letters indicate statistical significance groups at *p* < 0.05 (one-way ANOVA with post-hoc Tukey's multiple comparison test). The source data underlying the statistical analysis in **m–s** and uncropped images in **p–s** are provided in the Source Data file.

to the spikelet number per panicle and final grain yield[17]. Abundant molecular evidence has suggested that multiple regulators play essential roles in determining the fate of the reproductive meristem and in shaping panicle architecture associated with the primary branches[28–31], secondary branches[32–37], lateral spikelets[38–40], multifloret spikelets[41–43], and panicle type[44,45]. Mutants for the *Arabidopsis CLV3* ortholog *FON4* display enlarged panicles and an increased number of primary branches and floral organs, indicating that SSPs contribute to rice panicle morphogenesis[11–13]. Nevertheless, a limited number of SSPs and confirmed targeted receptors responsible for rice growth and development have been identified. To date, the specific small peptide ligand–receptor pairs required for rice panicle architecture are largely unknown. A previous study suggested that the OsER1–OsMKKK10–OsMKK4–OsMPK6 pathway regulates rice panicle morphogenesis by controlling cytokinin metabolism, demonstrating how upstream receptor signals maintain cytokinin homeostasis to shape plant inflorescence architecture[17]. However, the small peptide ligands recognized by the OsER1 receptor remained unknown.

In the present study, we first demonstrated that the small peptide ligands OsEPFL6, OsEPFL7, OsEPFL8, and OsEPFL9 (from the OsEPF/OsEPFL family) redundantly and synergistically contribute to rice panicle morphogenesis by recognizing the OsER1 receptor and subsequently activating the MAPK cascade. *OsEPFL6*, *OsEPFL7*, *OsEPFL8*, and *OsEPFL9* have negative regulatory effects on spikelet number per panicle, but *OsEPFL8* can especially control rice spikelet fertility (Fig. 1). By analyzing triple and quadruple null mutants for *OsEPFL6*, *OsEPFL7*, *OsEPFL8*, and *OsEPFL9*, we found that only the *osepfl6;7;9* triple mutant had significantly enhanced grain yield via increased spikelet number per panicle and maintaining normal spikelet fertility (Fig. 5j, k). Based on these findings, we propose a working model to describe the molecular mechanisms underlying optimization of rice panicle morphogenesis by suppressing these ligand–receptor pairs (Fig. 5l, m). During rice panicle morphogenesis, the active reproductive meristem produces the SSPs OsEPFL6, OsEPFL7, OsEPFL8, and OsEPFL9, which act as ligands for the OsER1 receptor. This triggers the OsMKKK10–OsMKK4–OsMPK6 cascade either directly or through unknown mediators. The activated OsMKKK10–OsMKK4–OsMPK6 cascade sequentially phosphorylates downstream substrates. However, the spatiotemporally activated dual-specificity phosphatase GRAIN SIZE AND NUMBER1 (GSN1) acts as a molecular "brake" to negatively regulate the OsER1–OsMKKK10–OsMKK4–OsMPK6 pathway; this occurs through inactivation of OsMPK6, which coordinately controls cell differentiation and proliferation in the panicle primordia and ultimately determines panicle architecture (Fig. 5l). However, in the panicle primordia of *osepfl6;7;9* triple mutants, the active reproductive meristem fails to produce OsEPFL6, OsEPFL7, and OsEPFL9; however, it does produce OsEPFL8, which recognizes the OsER1 receptor and attenuates the downstream OsMKKK10–OsMKK4–OsMPK6 cascade, altering the phosphorylation status of the substrates and the OsMPK6 phosphorylation regulatory network. These integrated signal outputs determine the spikelet

number per panicle and grain size, and thus optimize rice panicle architecture without affecting spikelet fertility, further enhancing grain yield (Fig. 5m). This is the first complete SSP–RLK regulatory model for rice panicle morphogenesis. It provides a framework for fundamentally understanding the role of ligand–receptor signaling in rice growth and development.

Based on the proposed working model, it is feasible to improve rice grain yield by shaping optimal panicle architecture through rational molecular design. In general, crop breeding is largely constrained by trade-offs between different agronomic traits. These include negative correlations among yield components, penalties in yield due to increased immunity, and decreased grain quality with increased yield[3,46,47]. Trade-offs among complex traits assist in maintaining relative fitness under unpredictable conditions and maximize reproductive success[48]. At present, only some cloned genes and weak alleles have been successfully used in rice breeding because many functional genes fail to overcome the trade-off effects, exposing the challenges in breeding by molecular design. Notably, in a previous study, we first uncovered the genetic basis for coordination of the trade-off between spikelet number per panicle and grain size in rice, which provided key insights into gene pleiotropy and the developmental plasticity of the panicle[3]. Although mitigating pleiotropy to overcome trade-off effects could be used in crop breeding, effective strategies currently available to achieve this goal are very limited. In the OsER1–OsMKKK10–OsMKK4–OsMPK6 pathway, loss of *OsER1* function increased spikelet number per panicle but compromised spikelet fertility, demonstrating the trade-off effect between spikelet abundance and fertility[17]. In the absence of optimal weak alleles of *OsER1* from natural variation, it is difficult to use this gene in genetic improvement of rice. This is a common problem in utilizing new functional genes for genetic crop improvement. Here, we bypassed the trade-off effect caused by the loss of OsER1 function by selectively removing its unique small peptide ligands, guaranteeing not only the specificity of signal perception but also the degree of signal activation. This genetic design is an effective strategy to overcome trade-offs between complex crop traits, embodying the concept of crop breeding by molecular design. In the future, it will be necessary to identify additional small peptide ligands of the OsER1 receptor to cultivate stress-tolerant and high-yield varieties.

In rice, there are 12 homologous members of the *OsEPF/OsEPFL* gene family: *OsEPF1*, *OsEPF2*, *OsEPFL1*, *OsEPFL2*, *OsEPFL3*, *OsEPFL4*, *OsEPFL5*, *OsEPFL6*, *OsEPFL7*, *OsEPFL8*, *OsEPFL9*, and *OsEPFL10* (Supplementary Fig. 1a). Although *OsEPF1*, *OsEPF2*, *OsEPFL1*, *OsEPFL2*, *OsEPFL3*, and *OsEPFL4* were found to be minimally expressed during young panicle development (Supplementary Fig. 2a), they could contribute to other physiological processes. A recent study reported that increasing *OsEPF1* and *OsEPF2* expression greatly reduced stomatal density in rice, indicating that *OsEPF1* and *OsEPF2* regulate stomatal development in a highly conserved way[19–21,49]. In the *aus* group variety Kasalath, loss-of-function of *OsEPFL2* rather than the *RAE2/GAD1*, led to short awn or awnless phenotype and reduced grain length[50], implying

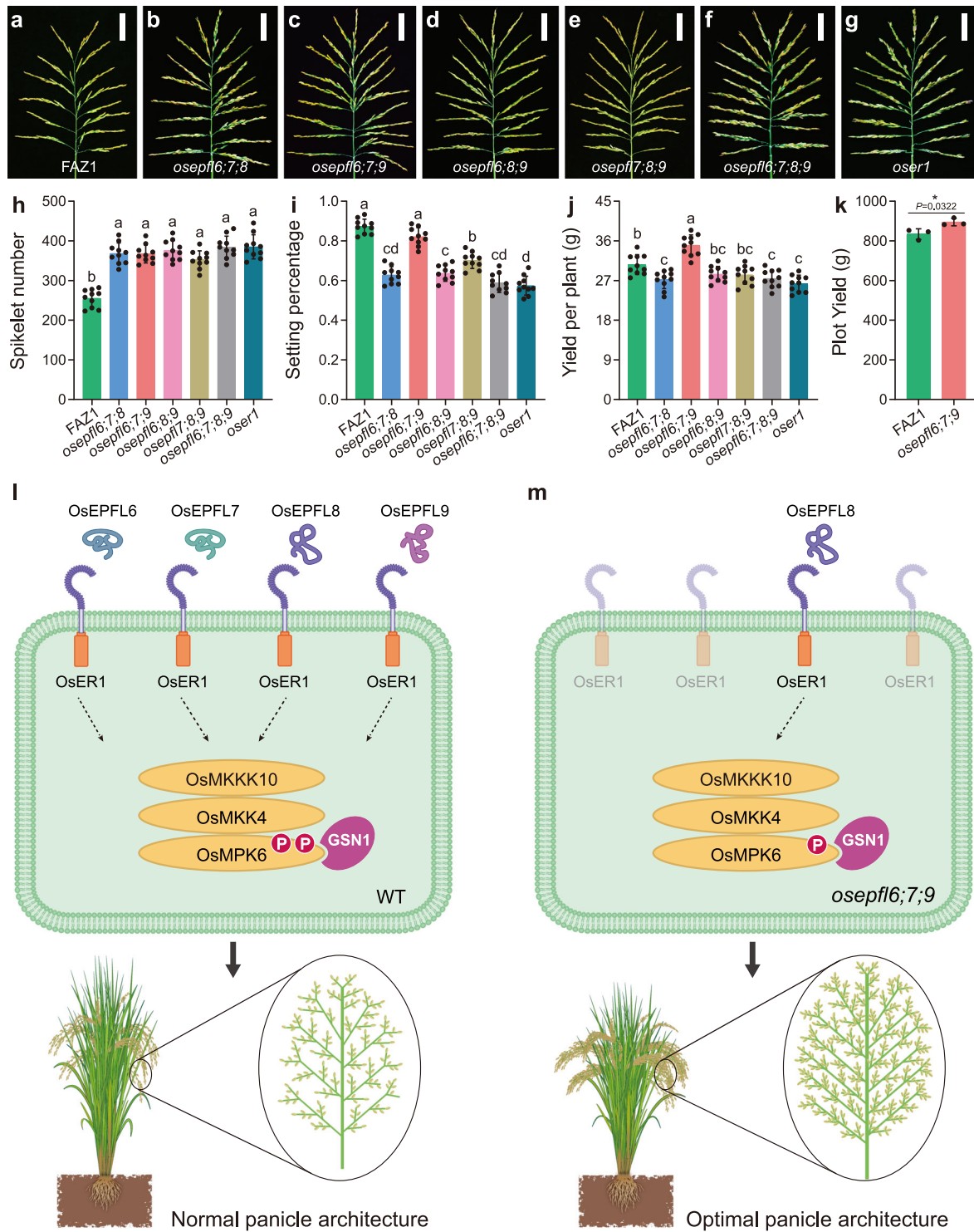

that members from the *OsEPF/OsEPFL* gene family may have been selected during rice domestication. Nevertheless, the functions of the *OsEPF/OsEPFL* gene family members are largely unknown, especially the corresponding specific ligand–receptor pairs. Interestingly, loss of *OsEPFL5* function resulted in decreased spikelet number per panicle but increased grain size, the opposite of the effect seen in *osepfl6*, *osepfl7*, *osepfl8*, and *osepfl9* null mutants (Fig. 1). Notably, OsMPK6 phosphorylation levels were clearly increased in the *osepfl5* mutant, in contrast to the *osepfl6*, *osepfl7*, *osepfl8*, and *osepfl9* mutants (Supplementary Fig. 10). This implied that the mature OsEPFL5 peptide may have an antagonistic role in the OsER1–MAPK pathway. In *Arabidopsis*,

the small peptide ligand EPFL9/Stomagen has been confirmed to promote stomatal development by recognizing ER family receptors; these receptors interfere with the inhibition of stomatal development by EPF1 and EPF2 through competitive binding to the ERf–TMM complex. This shows how a plant receptor agonist and antagonist use inhibitory and inductive cues to fine-tune tissue patterning on the plant epidermis[20–22]. It has not yet been demonstrated that OsEPFL5 can recognize OsER1, but we posit based on genetic evidence that mature OsEPFL5 is a potential ligand of the OsER1 receptor, implying that the competitive binding of antagonistic peptides could fine-tune rice panicle morphogenesis.

**Fig. 5 | Optimizing panicle architecture to enhance rice yield by suppressing ligand–receptor pairs. a–g** Rice panicles from FAZ1 (**a**) and the *osepfl6;7;8* (**b**), *osepfl6;7;9* (**c**), *osepfl6;8;9* (**d**), *osepfl7;8;9* (**e**), *osepfl6;7;8;9* (**f**), and *oser1* (**g**) mutants. Scale bar = 5 cm. **h–j** Comparison of average spikelet number per panicle (**h**), setting percentage (**i**), and yield per plant (**j**) between FAZ1, *osepfl6;7;8, osepfl6;7;9, osepfl6;8;9, osepfl7;8;9, osepfl6;7;8;9,* and *oser1*. Values are given as the mean ± SD (*n* = 10 plants). Different letters indicate statistical significance groups at *p* < 0.05 (one-way ANOVA with post-hoc Tukey's multiple comparison test). **k** Comparison of average plot yield between FAZ1 and *osepfl6;7;9*. Values are given as the mean ± SD (*n* = 3 plots). *\*p* < 0.05 indicates significant difference compared with FAZ1 by two-sided Student's *t* test. **l, m** Proposed working model for optimizing rice panicle architecture to improve grain yield by specifically suppressing ligand–receptor pairs. During young panicle morphogenesis, the active

reproductive meristem produces the small peptides OsEPFL6, OsEPFL7, OsEPFL8, and OsEPFL9, which act as ligands to recognize the OsER1 receptor. This triggers the OsMKKK10–OsMKK4–OsMPK6 cascade to phosphorylate downstream substrates. The spatiotemporally activated GSN1 acts as a molecular "brake" to negatively regulate the OsER1–OsMKKK10–OsMKK4–OsMPK6 pathway by inactivating OsMPK6, which controls rice panicle architecture (**l**). The active reproductive meristem of the *osepfl6;7;9* triple mutant failed to produce OsEPFL6, OsEPFL7, and OsEPFL9. Only OsEPFL8 could thus be perceived by OsER1, alleviating activation of the OsMKKK10–OsMKK4–OsMPK6 cascade and altering the phosphorylation status of the substrates. These integrated signal outputs thus generated optimized rice panicle architecture without affecting spikelet fertility, further enhancing grain yield (**m**). The source data underlying the statistical analysis in **h–k** are provided in the Source Data file.

The current data demonstrate that the small peptides OsEPFL6, OsEPFL7, OsEPFL8, and OsEPFL9 can act as ligands to recognize the OsER1 receptor; however, the mature OsEPFL8 and OsEPFL9 were found to have attenuated receptor affinity than OsEPFL6 and OsEPFL7 with ITC assays (Fig. 3a–e), which may represent either weaker affinity in vivo or poor folding of peptides expressed in insect cells. This also implies that the subtle amino acid differences could contribute to peptide processing or peptide folding and dimensional structure, and even the signal intensity and specificity. In addition, although the homologous *OsEPFL6, OsEPFL7, OsEPFL8,* and *OsEPFL9* have negative regulatory effects on spikelet number per panicle, *OsEPFL8* can especially control rice spikelet fertility (Fig. 1), thereby relating small peptides from the OsEPF/OsEPFL family to the floral organogenesis. In *Arabidopsis*, a new study reported that under low-temperature stress the *EPFL6* expressed in stamen filaments promotes filament elongation to ensure the alignment of stamen and pistil lengths and achieve successful self-pollination; however, at a moderate temperature, the entire *EPFL4/5/6* subfamily genes are required for the proper stamen-pistil growth coordination[51]. Recently, the *epfl1;2;3;4;5;6* sextuple mutant was found to show integument defects similar to both of the *er erl1;2* and *serk1;2;3* mutants, indicating that ERf–SERK-mediated EPFL signaling orchestrates the female gametogenesis and the development of surrounding integuments[52]. These results suggest that the redundant small peptide ligands and specific ligand–receptor pairs confer robustness to plant growth and against environmental stresses. SSPs have been found to play essential roles in numerous plant processes, including growth and development, responses to abiotic and biotic stresses, and beneficial microbial interactions[5]. Nonetheless, there are many SSPs in plant genomes that have not yet been discovered, which is largely due to a lack of effective methods and technologies. In the future, genomics, transcriptomics, proteomics, and new computational algorithms based on machine learning are expected to provide unprecedented capabilities for genome-wide identification of novel SSPs associated with plant development and environmental adaptation.

In conclusion, our findings reveal that the SSPs OsEPFL6, OsEPFL7, OsEPFL8, and OsEPFL9 act as ligands of the OsER1 receptor, activating the MAPK cascade to synergistically control rice panicle morphogenesis. These findings provide a framework for understanding how perceived ligand–receptor signals shape the rice panicle. Moreover, these findings provide significant insights into overcoming the trade-offs between complex traits and establish an innovative approach for breeding high-yield rice varieties by genetically manipulating the OsEPFL6–OsER1, OsEPFL7–OsER1, and OsEPFL9–OsER1 ligand–receptor pairs. However, little is known regarding the specific proteases that are required to process the precursors of the OsEPFL6, OsEPFL7, OsEPFL8, and OsEPFL9 small peptides that control rice panicle morphogenesis[14,53–55]. Additionally, the mechanisms by which small peptides are actively transported out of cells, whether through conventional or unconventional protein secretion pathways, remain elusive. In the

future, further discovery of components related to the OsEPFLs–OsER1–OsMKKK10–OsMKK4–OsMPK6 pathway through forward and reverse genetic analyses will unveil the detailed genetic basis of rice panicle morphogenesis and complex traits coupling.

## Methods

### Plant materials and growth conditions

All rice mutants were generated using CRISPR/Cas9 gene editing of the elite *indica* rice (*Oryza sativa*) variety FAZ1 and cross-fertilization. Rice plants were cultivated in experimental fields in Songjiang (Shanghai City, China) or Lingshui (Hainan Province, China) under natural growth conditions.

### Plasmid construction and plant transformation

The CRISPR/Cas9 system was used for gene editing of *OsEPFL6, OsEPFL7, OsEPFL8, OsEPFL9, OsEPFL5, OsER1,* and *OsER2*[27]. The designed sgRNAs were inserted into the pYLgRNA-OsU3 and pYLgRNA-OsU6a vector to produce sgRNA expression cassettes. Then, multiple sgRNA expression cassettes were cloned in the CRSPR/Cas9 binary vector to generate different pYLCRISPR/Cas9$_{ubi}$-H-OsU3-gRNA-OsU6a-gRNA constructs, respectively. To produce overexpression constructs, the full-length coding sequences of *OsEPFL6, OsEPFL7, OsEPFL8,* and *OsEPFL9* were each amplified from FAZ1 and separately cloned into the plant binary vector pCAMBIA1301 under the control of the ubiquitin promoter. The 2.5-kb regions upstream of the *OsER1, OsEPFL6, OsEPFL7, OsEPFL8,* and *OsEPFL9* start codons were amplified from FAZ1 and cloned into the pCAMBIA1300-GUSplus vector to generate the plasmids ProOsER1::GUS, ProOsEPFL6::GUS, ProOsEPFL7::GUS, ProOsEPFL8::GUS, and ProOsEPFL9::GUS. *Agrobacterium tumefaciens*-mediated transformation was conducted in rice using the strain EHA105. The DNA constructs used in this study were produced through seamless cloning with the NEBuilder HiFi DNA Assembly Master Mix (NEB) and confirmed by sequencing. The relevant PCR primers are shown in Supplementary Data 1.

### RNA extraction and qRT-PCR

Total plant RNA was extracted from individual rice tissues using Trizol reagent (Sangon Biotech, B511311). Reverse transcription (RT) was performed using ReverTra Ace qPCR RT Master Mix with gDNA Remover (TOYOBO, FSQ-301) and ~500 ng total RNA per sample. qRT-PCR was performed with the ABI 7300 Real-Time PCR System using Fast Start Universal SYBR Green Master Mix and ROX (Roche, 4913914001). *OsUBQ5* (LOC_Os01g22490) was used as the internal reference gene to normalize expression data using the $2^{-\Delta\Delta CT}$ method.

### GUS staining

Tissues of plants transformed with ProOsER1::GUS, ProOsEPFL6::GUS, ProOsEPFL7::GUS, ProOsEPFL8::GUS, and ProOsEPFL9::GUS were stained with standard method[56]. In brief, samples from rice in the reproductive stage were immersed and vacuum-infiltrated with GUS staining buffer (1 mM X-Gluc, 50 mM phosphate buffer at pH 7.0,

0.4 mM each $K_3Fe(CN)_6$/$K_4Fe(CN)_6$, 0.1% Triton X-100), then incubated at 37 °C overnight. Samples were cleared in 75% ethanol to remove chlorophyll, then photographed with a Leica S9D stereomicroscope.

## Protein expression and purification
The coding sequences of the extracellular LRR domains of OsER1 (residues 25–577), OsEPFL6 (residues 72–122), OsEPFL7 (residues 91–141), OsEPFL8 (residues 92–142), and OsEPFL9 (residues 88–138) were cloned into modified pFastBac vectors containing an N-terminal hemolin (Hem) signal peptide and a cleavable N-terminal 6× His-SUMO tag (Hem SUMO), with or without a C-terminal 6×His tag. With the Bac-to-Bac baculovirus expression system (Invitrogen) following the manufacturer's protocol, all proteins were expressed in High Five cells at 22 °C. High Five cells were grown in ESF 921 medium (Expression Systems, 96001) with 120 rpm shaking at 28 °C until the density reached $2\times10^6$ cells/mL. One liter of cells ($2\times10^6$ cells/mL) was infected with 20 mL of recombinant baculovirus. The supernatant was harvested by centrifugation 60 h after infection. The supernatant was applied to a Ni-NTA column (Novagen, 70666). Bound proteins were eluted in buffer containing 25 mM Tris (pH 8.0), 150 mM NaCl, and 250 mM imidazole. Proteins were further purified by size exclusion chromatography (HiLoad 16/600 Superdex 200 prep grade, GE Healthcare, 28989335) in a buffer containing 10 mM Bis-Tris (pH 6.0) and 100 mM NaCl. The relevant PCR primers used for cloning the relevant gene fragments are listed in Supplementary Data 1.

## ITC assay
Binding affinity of OsER1$^{LRR}$ for OsEPFL6, OsEPFL7, OsEPFL8, and OsEPFL9 was measured using an ITC200 (MicroCal LLC) at 25 °C in a buffer containing 10 mM Bis-Tris (pH 6.0) and 100 mM NaCl. Approximately 0.3 mM OsEPFL6, OsEPFL7, OsEPFL8, or OsEPFL9 small peptides were injected into the stirred calorimeter cell (250 μL) containing 0.03 mM OsER1$^{LRR}$ with $24\times1.5$ μL at intervals of 150 s. The stirring speed was 750 rpm. Measurements of the binding affinity based on the titration data were analyzed using the MicroCal origin software.

## Gel filtration assay
OsER1$^{LRR}$ and the OsEPFL6, OsEPFL7, OsEPFL8, and OsEPFL9 small peptides were purified as described above, then mixed and incubated for 30 min at 4 °C. They were then analyzed via gel filtration (HiLoad 16/600 Superdex 200 prep grade, GE Healthcare, 28989335) in a buffer containing 10 mM Bis-Tris (pH 6.0) and 100 mM NaCl. Samples from relevant fractions were further separated with SDS-PAGE and visualized with Coomassie brilliant blue staining.

## Co-IP
The coding sequence of the LRR domain of OsER1 was cloned into the pCAMBIA1306-Flag (3×) plasmid to generate the Pro35S::OsER1$^{LRR}$-Flag vector. The coding sequences of OsEPFL6, OsEPFL7, OsEPFL8, and OsEPFL9 were each cloned into the pCAMBIA1301-Myc (7×)-His (6×) plasmid to produce the Pro35S::OsEPFL6-Myc, Pro35S::OsEPFL7-Myc, Pro35S::OsEPFL8-Myc, and Pro35S::OsEPFL9-Myc vectors. Pro35S::OsER1$^{LRR}$-Flag was then transiently co-expressed with Pro35S::OsEPFL6-Myc, Pro35S::OsEPFL7-Myc, Pro35S::OsEPFL8-Myc, or Pro35S::OsEPFL9-Myc in N. benthamiana leaf cells using A. tumefaciens strain GV3101. Membrane proteins were extracted with the modified protocol[19]. Rice young panicles were ground to fine powder and solubilized with extraction buffer (100 mM Tris-HCl at pH 8.8, 150 mM NaCl, 1 mM EDTA, 20% glycerol, 20 mM NaF, 1 mM PMSF). The extracts were then sonicated on ice and ultracentrifuged at 130000 g for 30 min at 4 °C to obtain the membrane fraction as precipitate, which was resuspended in membrane solubilization buffer (100 mM Tris-HCl at pH 7.5, 150 mM NaCl, 1 mM EDTA, 10% glycerol, 1% Triton X-100, 20 mM NaF, 1 mM PMSF) to release membrane proteins. Anti-Flag M2 agarose beads (Sigma-Aldrich, A2220) were used to perform the Co-IP assays. Immunoblot assays were performed using anti-Flag (CST, 14793) and anti-Myc (CST, 2276) antibodies. The relevant PCR primers used for cloning the relevant gene fragments are listed in Supplementary Data 1.

## Protein extraction and immunoblot assay
Protein extraction and immunoblot assays were performed as modified method below[3]. Rice young panicle samples were harvested and soluble proteins were extracted with a Plant Total Protein Extraction Kit (Sigma-Aldrich, PE0230). Briefly, pooled 1 mm to 3 mm young panicles were ground to a fine power in liquid nitrogen, then the powder was rinsed with methanol followed by acetone. The supernatant was removed from each sample, and the remaining tissue was then dried and dissolved. Proteins were denatured by adding concentrated SDS loading buffer and boiling for 5 min, then were separated on a precast 4-12% Bis-Tris gel (Tanon, 1808008H). Endogenous OsMPK6 levels were assayed via immunoblot using anti-OsMPK6 antibody at 1:3000 dilution (Sigma-Aldrich, A7104). Phosphorylated OsMPK6 was visualized using an optimized immunoblot analysis with immunoreaction enhancer solutions (TOYOBO, NKB-101) and anti-Phos-OsMPK6 antibody at 1:1000 dilution (CST, 4370), which specifically recognizes the conserved dual-phosphorylated T-E-Y motif of phosphorylated OsMPK6. The loading control was probed using anti-Actin antibody at 1:5000 dilution. (Abmart, M20009)

## Statistical analysis
For phenotype analysis, qRT-PCR analysis, and protein quantification, statistical analysis was assessed as described in the figure legends. Significant differences between two groups were determined with two-sided Student's $t$ test; differences between three or more groups were determined using one-way analysis of variance (ANOVA) with post-hoc Tukey's multiple comparison test. All analyses were performed using GraphPad Prism 8 software and were shown in the graphs or source data.

## Accession numbers
Gene sequence data from this article can be found in the MSU Rice Genome Annotation Project Database under the following accession numbers: OsEPFL6, LOC_Os03g06610; OsEPFL7, LOC_Os11g37190; OsEPFL8, LOC_Os05g39880; OsEPFL9, LOC_Os01g60900; OsEPFL5, LOC_Os07g04020; OsER1, LOC_Os06g10230; and OsER2, LOC_Os06g03970.

## Reporting summary
Further information on research design is available in the Nature Portfolio Reporting Summary linked to this article.

# Data availability
The data that support this study are present in the paper and its Supplementary Information files. The genetic materials generated and analyzed during the current study are available from the corresponding author upon reasonable request. Source data are provided in this paper.

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

## Acknowledgements

We thank Dr. Yaoguang Liu (South China Agriculture University) for the donation of CRISPR/Cas9 plasmids. This work was supported by grants from the National Key Research and Development Program of China (2022YFD1200102), Chinese Academy of Sciences (XDB27010104, 159231KYSB20200008), National Natural Science Foundation of China (31788103), Laboratory of Lingnan Modern Agriculture Project (NT2021002), CAS-Croucher Funding Scheme for Joint Laboratories, China Postdoctoral Science Foundation (2020T130673) and National Key Laboratory of Plant Molecular Genetics.

## Author contributions

H.X.L. and J.C. conceived and supervised the project, and T.G., Z.Q.L., Y.X., H.X.L., and J.C. designed the experiments. T.G., Z.Q.L., and Y.X. performed most of the experiments. J.X.S, W.W.Y., N.Q.D., Y.K., Y.B.Y., H.Y.Z., H.X.Y., S.Q.G., J.J.L., B.L., and H.X.L. performed some of the experiments. T.G., H.X.L., and J.C. analyzed data and wrote the manuscript.

## Competing interests

The authors declare no competing interests.
