## [Peer Review File · Nature Communications]

Optimization of rice panicle architecture by specifically suppressing ligand–receptor pairsReviewer #1 (Remarks to the Author):

Rice panicle architecture is an important agronomic trait. The authors previously reported that the OsER1-OsMCKK10-OsMCKK4-OsMPK6 pathway shapes panicle architecture by regulating cytokinin metabolism. This manuscript further demonstrates that the OsEPF/OsEPFL small peptide family members (OsEPFL6, OsEPFL7, OsEPFL8, and OsEPFL9) act as ligands of the OsER1 receptor to regulate panicle architecture by activating the OsMCKK10-OsMCKK4-OsMPK6 cascade. Similarly, the EPF/EPFL-ERF-MAPK signaling pathway has been reported to regulate stomatal development and organ growth in Arabidopsis. OsEPFL6, OsEPFL7, OsEPFL8, and OsEPFL9 negatively regulate spikelet number per panicle, and OsEPFL8 also controls rice spikelet fertility. The *osepfl6 osepfl7 osepfl9* triple mutant enhanced grain yield without affecting spikelet fertility. In addition, OsEPFL5 acts as a positive regulator of spikelet number. Thus, this study identifies four OsEPF/OsEPFL peptides as ligands of OsER1, revealing important mechanisms underlying panicle architecture regulation. Addressing the issues mentioned below may help to improve this manuscript.

1. The authors generated *epfl* mutants using the genome editing technology, and double and triple mutant analyses revealed the complicated genetic interactions with respect to different traits (e.g. spikelet number, grain size and plant height). For each EPFL gene, only one allele was generated for phenotypic analysis. For example, the *osepfl8* allele showed the increased spikelet number, but the decreased fertility. As only one *osepfl8* allele was analyzed, it is not convincing. The genome editing technology and the rice transformation process could cause multiple mutations in the genome. It will be better to generate multiple alleles or conduct genomic complementation test to confirm the phenotypes of these *epfl* mutants.

2. OsEPFL5 positively regulates spikelet number per panicle, while OsEPFL6, OsEPFL7, OsEPFL8, and OsEPFL9 negatively regulate spikelet number per panicle. The authors proposed that OsEPFL5 is not the ligand of OsER1. I speculate that OsEPFL5 may compete with OsEPFL6, OsEPFL7, OsEPFL8, and OsEPFL9 to bind OsER1. The protein interaction between OsEPFL5 and OsER1 should be tested, and competitive binding assay should also be performed.

3. It is important that *osepfl6 osepfl7 osepfl9* triple mutant enhanced grain yield per plant. It will be better to investigate grain yield per plot. The yield is mainly determined by tiller number/panicle number, spikelet number and grain weight. However, the authors only measured spikelet number and grain size of these mutants. In fact, tiller number plays a predominant role in determining rice grain yield. It is unclear whether these mutations affect the tiller number. It will help understand whether the increased spikelet number or tiller number is mainly responsible for high yield in *osepfl6 osepfl7 osepfl9* triple mutant.

4. The levels of phos-MAPK6 in *oser*, *osepfl* mutants and overexpression lines were investigated. The levels of phos-MAPK6 in several mutants are not fully consistent with their phenotypes. For example, the level of phos-MAPK6 looks higher in *oser1* than that in *osepfl6 osepfl8*, while the spikelet number phenotype of *oser1* is stronger than that of *osepfl6 osepfl8* (Figure 4). It is necessary to quantify the levels of phos-MAPK6 with biological replicates in main figures.

Reviewer #2 (Remarks to the Author):

The manuscript by Guo et al. describes the role of four genes, OsEPFL6, OsEPFL7, OsEPFL8, and OsEPFL9, in architecture and fecundity of rice plants. Previously this group uncovered that a signaling pathway consisting of receptor kinase ERECTA1 (OsER1) and a downstream MAP kinase cascade controls rice panicle morphogenesis. It reduces number of spikelets per panicle and therefore reduces seed yield. However, a knockout of OsER1 has pleiotropic effects: while it increases spikelet number, it also reduces spikelet fertility. In this work, authors went after potential ligands of OsER1 with the goal of finding a mutant with increased spikelet number but without reduced fertility or decreased seed size. Based on work in Arabidopsis, small proteins from EPF/EPFL family function as ligands for ERECTA family receptors. Authors performed detailed

phenotypic analysis of single and higher order mutants of five OsEPFL genes and identified that the *osepfl6 osepfl7 osepfl9* mutant forms more fertile spikelets, but slightly smaller seeds compared to wild type. Overall, the grain yield per plant is increased in that mutant. These findings open a new avenue for optimization of rice architecture to achieve higher yields and are of high significance for agriculture. The authors also investigate whether the five OsEPFLs are truly ligands of ER1 and whether they can alter activity of the MAP kinase cascade. Predictably the answer is yes. This work is useful but less exciting as it mostly confirms that this pathway function similarly in rice and other plant species. I wish they done instead the study of changes to the structure of inflorescence meristem and branch primordia in *osepfl* mutants to uncover reasons behind altered structures of panicles. However, current work already has a lot of interesting and significant findings, and this study can be done in the future. Overall, the conclusions are well supported by the data and the manuscript is well written. The figures are not well organized but that is easy to fix (see recommendations below).

Specific recommendations:

1. The manuscript does not organize data in a straightforward way. There are a lot of bar graphs that are difficult to grasp. Often the same data is unnecessary shown in multiple figures. The use of color in bar graphs is not consistent.
 - a. Fig. 2 Data in panels m- t is repetitive. E.g. the same data for FAZ and *oser1* is repeated 8 times. Create only one panel for m-t. Do not present the same data multiple times in the same figure. Remove data about single mutants from that figure as you already presented that data in Fig. 1.
 - b. Fig 5. Data in panels h-k and i-o is repetitive. Create one graph for spikelet number and one for yield per plant. Here is no need to have so many panels.
 - c. Extended data Fig 5. Combine panels a-h in one panel and i-p in a different panel. Do not present the same data multiple times.
 - d. Extended data Fig 6. Combine panels a-h in one panel and i-p in a different panel.
 - e. Extended data Fig 7. Combine panels m-t in one panel.
 - f. Extended data Fig 8. Combine panels a-h in one panel and i-p in a different panel.
 - g. Extended data Fig 14. Combine panels a through d, e through h, I through l, m through p, g through t.
 - h. Extended data Fig 15. Combine panels o through r, s through v.
 - i. Fig 4. Color use in graphs m-o is not consistent. While in n and o the bars of the same color belong to the same genotype, in m the colors belong to different genotypes. This is confusing. Change colors in m. In general, make sure colors are used for a purpose and not randomly assigned. In a single figure do not use the same color for different genotypes. If each genotype described only by one data set in a figure, the same color (gray) can be used for all bars.
 - j. Fig 5p. Make all OsER1 molecules the same color. It is not clear why they colored differently. It is not clear what the different color of substrates in p and q represent. As current data does not describe the substrates, I would remove substrates from that figure altogether. I also recommend removing P from OsMKK4 as you did not study it. Current image suggests that phosphorylation of OsMKK4 is not changed in wt versus *osepfl6,7,8*.
2. Extended data Fig. 11k; The data in the top panel is of poor quality and has to be redone or removed.
3. Line 418. It is not obvious based on the presented Western blot that phosphorylation of OsMPK6 is increased. Statistical analysis of multiple Western blots is recommended.
4. Fig 3a-e. The affinity of OsEPFL8 and OsEPFL9 for OsER1 is rather low. Does this represent reduced affinity in vivo or is it due to poor folding of peptides expressed in insect cells? Are these peptides functional? Please discuss in the manuscript.
5. Line 128. Change ER1 to ER. There is no ER1 gene in Arabidopsis.
6. In introduction and discussion, the paper is focused too narrowly on ER/EPFL signaling pathway in rice. This pathway has been extensively studied in other species especially in Arabidopsis. Please more broadly discuss and cite relevant data from other species. For example, discuss the function of Arabidopsis genes homologous to OsEPFL5,6,7,8,9 and compared to function of these genes in rice and in Arabidopsis.
7. Line 531 Remove word "ingenious". This is overstatement.

Point-by-point Response to Reviewers

Dear Reviewers:

We are very grateful for your recognitions and comments concerning our manuscript. You provided really valuable insights and thoughtful suggestions, which are all very helpful for improving our manuscript and guiding our future research. Based on your suggestions, we have made careful modifications to the original manuscript. We hope that the revised manuscript is more acceptable and satisfactory. Below you will find our point-by-point responses to the reviewers' comments:

Reviewer #1 (Remarks to the Author):

Rice panicle architecture is an important agronomic trait. The authors previously reported that the OsER1-OsMKKK10-OsMKK4-OsMPK6 pathway shapes panicle architecture by regulating cytokinin metabolism. This manuscript further demonstrates that the OsEPF/OsEPFL small peptide family members (OsEPFL6, OsEPFL7, OsEPFL8, and OsEPFL9) act as ligands of the OsER1 receptor to regulate panicle architecture by activating the OsMKKK10-OsMKK4-OsMPK6 cascade. Similarly, the EPF/EPFL-ERf-MAPK signaling pathway has been reported to regulate stomatal development and organ growth in Arabidopsis. OsEPFL6, OsEPFL7, OsEPFL8, and OsEPFL9 negatively regulate spikelet number per panicle, and OsEPFL8 also controls rice spikelet fertility. The *osepfl6 osepfl7 osepfl9* triple mutant enhanced grain yield without affecting spikelet fertility. In addition, OsEPFL5 acts as a positive regulator of spikelet number. Thus, this study identifies four OsEPF/OsEPFL peptides as ligands of OsER1, revealing important mechanisms underlying panicle architecture regulation. Addressing the issues mentioned below may help to improve this manuscript.

1. The authors generated *epfl* mutants using the genome editing technology, and double and triple mutant analyses revealed the complicated genetic interactions with

respect to different traits (e.g. spikelet number, grain size and plant height). For each EPFL gene, only one allele was generated for phenotypic analysis. For example, the *osepfl8* allele showed the increased spikelet number, but the decreased fertility. As only one *osepfl8* allele was analyzed, it is not convincing. The genome editing technology and the rice transformation process could cause multiple mutations in the genome. It will be better to generate multiple alleles or conduct genomic complementation test to confirm the phenotypes of these *epfl* mutants.

Response :

Thank you for your good comments and sincere suggestions. In our study, we used genome editing technology to generate multiple mutant alleles for the *OsEPFL6*, *OsEPFL7*, *OsEPFL8*, *OsEPFL9*, and *OsEPFL5*, respectively. According your comments, in the revised manuscript, we now have added new mutant alleles for the *OsEPFL6*, *OsEPFL7*, *OsEPFL8*, *OsEPFL9*, and *OsEPFL5* to confirm their phenotypes (new Supplementary Fig. 4, Fig. 5). However, in the process of producing the triple and quadruple mutants, only a kind of specific higher-order genotype were screened and the phenotypes were observed over multiple generations. Therefore, in the manuscript only the corresponding single mutants were showed. Although we didn't conduct genomic complementation test to confirm the phenotypes of these mutants, we purified the mutants by backcross and observed the phenotypes over multiple generations. We hope the reviewer could understand. Thank you very much.

2. *OsEPFL5* positively regulates spikelet number per panicle, while *OsEPFL6*, *OsEPFL7*, *OsEPFL8*, and *OsEPFL9* negatively regulate spikelet number per panicle. The authors proposed that *OsEPFL5* is not the ligand of *OsER1*. I speculate that *OsEPFL5* may compete with *OsEPFL6*, *OsEPFL7*, *OsEPFL8*, and *OsEPFL9* to bind *OsER1*. The protein interaction between *OsEPFL5* and *OsER1* should be tested, and competitive binding assay should also be performed.

Response :

We thank the reviewer for these good and constructive comments. Actually, we have tried to express and purify the mature *OsEPFL5* peptide for many times.

Unfortunately, we failed to express this small peptide *in vitro* due to the distinctive complex structure possibly (Supplementary Fig. 1b, c). This is a very big hurdle to test the protein interaction between OsEPFL5 and OsER1, even the competitive binding assay. Therefore, we only proposed this idea: “It has not yet been demonstrated that OsEPFL5 can recognize OsER1, but we posit based on genetic evidence that mature OsEPFL5 is a potential ligand of the OsER1 receptor” in the discussion part (Please see the lines 574-576). We look forward to solving the issues in the future. We hope the reviewer could approve this. Thank you for your insightful comments again.

3. It is important that osepf16 osepf17 osepf19 triple mutant enhanced grain yield per plant. It will be better to investigate grain yield per plot. The yield is mainly determined by tiller number/panicle number, spikelet number and grain weight. However, the authors only measured spikelet number and grain size of these mutants. In fact, tiller number plays a predominant role in determining rice grain yield. It is unclear whether these mutations affect the tiller number. It will help understand whether the increased spikelet number or tiller number is mainly responsible for high yield in osepf16 osepf17 osepf19 triple mutant.

Response :

Thank you for your good suggestions. We have investigated the grain yield per plot this summer (new Fig. 5k), and the data have been described in the revised manuscript (Line 459-460). As you mentioned, the rice grain yield is mainly determined by tiller number, spikelet number and grain weight, and the tiller number plays a predominant role in determining grain yield. In fact, we found that these mutations don't affect the rice tiller number, which is consistent with the *oser1* mutant. We have showed the tiller number data in the new Supplementary Fig. 15q, and also described in the revised manuscript (Lines 464-465). Thanks a lot.

4. The levels of phos-MAPK6 in *oser*, *osepf1* mutants and overexpression lines were investigated. The levels of phos-MAPK6 in several mutants are not fully consistent

with their phenotypes. For example, the level of phos-MAPK6 looks higher in *oser1* than that in *osepfl6 osepfl8*, while the spikelet number phenotype of *oser1* is stronger than that of *osepfl6 osepfl8* (Figure 4). It is necessary to quantify the levels of phos-MAPK6 with biological replicates in main figures.

Response :

Thank you very much for your good comments. The phosphorylation level of OsMPK6 is really associated with the rice panicle development. However, it was somewhat difficult to assay the phosphorylation level of OsMPK6 to make it relatively stable using the young panicle from different mutants because we couldn't ensure that the young panicles were at exactly same growth period. Moreover, it is difficult to absolutely correlate the phosphorylation level of OsMPK6 to the single trait such as spikelet number per panicle or grain size among the double and triple mutants and *oser1* mutant, because multiple developmental processes occurred simultaneously in the young panicle period. Nevertheless, the phosphorylation level of OsMPK6 was obviously reduced in the multiple mutants compared with FAZ1. We also noticed that the phosphorylation level of OsMPK6 looks somewhat higher in *oser1* than that in *osepfl6;8* double mutant in original Figure 4. To avoid misunderstanding, we have replaced the previous figure with the new result (Fig. 4p). We thereby repeated the assays and quantified the phosphorylation level of OsMPK6 with biological replicates in the figures according to your suggestion. The results have been showed below the panels (Fig. 2o-r, Fig. 4p-s and Supplementary Fig. 10, Fig. 13i, Fig. 16). We hope they are approved. Thank you again.

Reviewer #2 (Remarks to the Author):

The manuscript by Guo et al. describes the role of four genes, OsEPFL6, OsEPFL7, OsEPFL8, and OsEPFL9, in architecture and fecundity of rice plants. Previously this group uncovered that a signaling pathway consisting of receptor kinase ERECTA1 (OsER1) and a downstream MAP kinase cascade controls rice panicle morphogenesis. It reduces number of spikelets per panicle and therefore reduces seed yield. However, a knockout of OsER1 has pleiotropic effects: while it increases spikelet number, it also reduces spikelet fertility. In this work, authors went after potential ligands of OsER1 with the goal of finding a mutant with increased spikelet number but without reduced fertility or decreased seed size. Based on work in Arabidopsis, small proteins from EPF/EPFL family function as ligands for ERECTA family receptors. Authors performed detailed phenotypic analysis of single and higher order mutants of five OsEPFL genes and identified that the *osepfl6 osepfl7 osepfl9* mutant forms more fertile spikelets, but slightly smaller seeds compared to wild type. Overall, the grain yield per plant is increased in that mutant. These findings open a new avenue for optimization of rice architecture to achieve higher yields and are of high significance for agriculture. The authors also investigate whether the five OsEPFLs are truly ligands of ER1 and whether they can alter activity of the MAP kinase cascade. Predictably the answer is yes. This work is useful but less exciting as it mostly confirms that this pathway function similarly in rice and other plant species. I wish they done instead the study of changes to the structure of inflorescence meristem and branch primordia in *osepfl* mutants to uncover reasons behind altered structures of panicles. However, current work already has a lot of interesting and significant findings, and this study can be done in the future. Overall, the conclusions are well supported by the data and the manuscript is well written. The figures are not well organized but that is easy to fix (see recommendations below).

Response :

We thank the reviewer for providing these excellent comments and sincere suggestions to guide our future research. We have revised the original manuscript respectively. We supposed that the revised version has improved a lot with your

suggestions.

Specific recommendations:

1. The manuscript does not organize data in a straightforward way. There are a lot of bar graphs that are difficult to grasp. Often the same data is unnecessary shown in multiple figures. The use of color in bar graphs is not consistent.

a. Fig. 2 Data in panels m- t is repetitive. E.g. the same data for FAZ and oser1 is repeated 8 times. Create only one panel for m-t. Do not present the same data multiple times in the same figure. Remove data about single mutants from that figure as you already presented that data in Fig. 1.

b. Fig 5. Data in panels h-k and i-o is repetitive. Create one graph for spikelet number and one for yield per plant. Here is no need to have so many panels.

c. Extended data Fig 5. Combine panels a-h in one panel and i-p in a different panel. Do not present the same data multiple times.

d. Extended data Fig 6. Combine panels a-h in one panel and i-p in a different panel.

e. Extended data Fig 7. Combine panels m-t in one panel.

f. Extended data Fig 8. Combine panels a-h in one panel and i-p in a different panel.

g. Extended data Fig 14. Combine panels a through d, e through h, I through l, m through p, g through t.

h. Extended data Fig 15. Combine panels o through r, s through v.

i. Fig 4. Color use in graphs m-o is not consistent. While in n and o the bars of the same color belong to the same genotype, in m the colors belong to different genotypes. This is confusing. Change colors in m. In general, make sure colors are used for a purpose and not randomly assigned. In a single figure do not use the same color for different genotypes. If each genotype described only by one data set in a figure, the same color (gray) can be used for all bars.

j. Fig 5p. Make all OsER1 molecules the same color. It is not clear why they colored differently. It is not clear what the different color of substrates in p and q represent. As current data does not describe the substrates, I would remove substrates from that figure altogether. I also recommend removing P from OsMKK4 as you did not study it.

Current image suggests that phosphorylation of OsMKK4 is not changed in wt versus osepfl6,7,8.

Response :

Thank you for your good and careful suggestions. We have reorganized some data in a straightforward way, especially the bar graphs and the data arrangement.

- a. According to your suggestion, we have created only one panel for Fig. 2 m-t in the revised Fig. 2 m; however, in order to facilitate comparisons, we reserved the data about single mutants (Fig. 2m, n).
- b. We have also created one graph for spikelet number and one for yield per plant in the revised Fig. 5 h-j (Fig. 5h-j).
- c. We have revised this figure with one graph for each trait. Please see the new figure (Supplementary Fig. 7a, b).
- d. We have revised this figure with one graph for each trait. Please see the new figure (Supplementary Fig. 7c, d).
- e. We have revised this figure with one graph for plant height. Please see the new figure (Supplementary Fig. 8m).
- f. We have revised this figure with one graph for each trait. Please see the new figure (Fig. 2n and Supplementary Fig. 7e).
- g. We have revised this figure with one graph for each trait. Please see the new figure (Fig. 5h-j and Supplementary Fig. 14a-d).
- h. We have revised this figure with one graph for plant height and stem diameter. Please see the new figure (Supplementary Fig. 15o, p).
- i. Thank you for this good suggestion. We have modulated the bar color on the basis of different genotypes for the Fig. 4 (Fig. 4m-o).
- j. Thank you very much. We have revised the proposed model referring to your comments. Please see the new figure (Fig. 5l, m).

2. Extended data Fig. 11k; The data in the top panel is of poor quality and has to be redone or removed.

Response :

Thank you for your sincere suggestion. We are sorry it was somewhat obscure. In fact, it was very difficult to assay the phosphorylation level of OsMPK6 in the rice seedlings. We have been trying for many times. Although it looks poor quality, the data also suggest that the mature OsEPFL6 small peptide promotes shoot elongation depending on the OsER1–MAPK pathway. Here, we decide to remove this figure to avoid misunderstandings (Supplementary Fig. 11). We hope that the reviewer will agree this. Thank you again.

3. Line 418. It is not obvious based on the presented Western blot that phosphorylation of OsMPK6 is increased. Statistical analysis of multiple Western blots is recommended.

Response :

Thank you for your good suggestion. We are sorry that we failed to assay a very high phosphorylation level of OsMPK6 in the overexpression lines possibly owe to our sampling period or self-feedback mechanism of the overexpression plants. As you recommended, we repeated the assays and quantified the phosphorylation level of OsMPK6 with biological replicates in the figures. These results have been showed below the panels (Supplementary Fig. 13i), and indicated that OsMPK6 phosphorylation levels were also significantly increased in *OsEPFL7*- and *OsEPFL9*-overexpression lines compared with FAZ1. We hope they are approved. Thank you again.

4. Fig 3a-e. The affinity of OsEPFL8 and OsEPFL9 for OsER1 is rather low. Does this represent reduced affinity in vivo or is it due to poor folding of peptides expressed in insect cells? Are these peptides functional? Please discuss in the manuscript.

Response :

Thank you for your concern. Our ITC data suggested that the affinity of OsEPFL8 and OsEPFL9 for OsER1 is relatively low. We supposed that the affinity difference between OsEPFL6/7 and OsEPFL8/9 is due to the amino acid composition of small

peptide. Although we found that the OsEPFL6/7 and OsEPFL8/9 are very homologous in the essential amino acid residues (Supplementary Fig. 1b), maybe the peptide processing or peptide folding and dimensional structure have some specific differences which contributes to their recognition to OsER1 receptor. Thereby, we supposed that the lower affinity of OsEPFL8 and OsEPFL9 for OsER1 could represent weaker affinity *in vivo* rather than poor folding of peptides expressed in insect cells. Furthermore, in the signal transduction pathway, the lower affinity and higher affinity per se could represent different signal intensity and specificity. In this study, the mature OsEPFL6 small peptide was chosen as a representative to determine whether it could promote rice seedling growth. We found that different concentrations of the mature OsEPFL6 small peptide could promote elongation of the shoot; however, it failed in the *oser1* mutant, which confirmed the physiological activity and function of the purified small peptides (Supplementary Fig. 11). Thereby, we supposed that these small peptides purified in insect cells are functional. We are sorry that we didn't test each small peptide purified from insect cells because it is really hard to get enough small peptides to treat plants. According to your comments, we have discussed this part (described in the revised manuscript lines 579-598). We hope that the reviewer will agree this. Thank you very much.

5. Line 128. Change ER1 to ER. There is no ER1 gene in Arabidopsis.

Response :

Thank you very much. We have changed ER1 to ER. Please see the revised manuscript (Line 134).

6. In introduction and discussion, the paper is focused too narrowly on ER/EPFL signaling pathway in rice. This pathway has been extensively studied in other species especially in Arabidopsis. Please more broadly discuss and cite relevant data from other species. For example, discuss the function of Arabidopsis genes homologous to OsEPFL5,6,7,8,9 and compared to function of these genes in rice and in Arabidopsis.

Response :

Thank you for your good suggestions. We are sorry that we lost sight of relevant data from other species in original manuscript. We mainly focused on ER/EPFL signaling pathway in rice panicle morphogenesis. In the revised manuscript, we have cited relevant references (new references 24, 25, 26, 50, 51, and 52) and discussed more broadly. According to your suggestion, we also discussed and compared the function of homologous genes in rice and *Arabidopsis* (described in the revised manuscript lines 91-97, lines 558-561, and lines 589-598). Many thanks.

7. Line 531 Remove word “ingenious”. This is overstatement.

Response :

Thank you for your suggestion. We are sorry that it was overstated. We have removed this word in the revised version. Please see the revised sentence (Lines 545-547).

Reviewer #1 (Remarks to the Author):

The authors have addressed my main concerns in this revision. It is a nice paper.

Reviewer #2 (Remarks to the Author):

Guo et al. satisfactorily addressed most of my concerns in this revision. The new display of data in the figures is more straightforward and easier to understand. The added statistic for Western blots builds confidence in their conclusions. One concern left remaining is that their data does not distinguish whether OsEPFL8 and OsEPFL9 proteins isolated from insect cells are correctly folded and suitable for ITC or whether OsER1 is not their main receptor. This refers to Fig 3c and d; item 4 in the previous review. The Authors discuss this point in the paper and in their rebuttal but in a rather confusing manner. The fact that OsEPFL6 isolated from insect cells is functional does not guarantee that other OsEPFLs isolated from insect cells are functional. Based on our experience working with small plant Cys -rich peptides isolated from insect cells, some peptides do not fold correctly, and as the result have low affinity for receptors and are physiologically inactive. Other peptides from the same gene family fold correctly and are functional in vivo. Based on the data in the manuscript it is not clear whether the OsEPFL8 and 9 peptides are of high quality and the data obtained in the ITC experiment is reliable. However, since affinity of OsEPFL6,7,8,9 for OsER1 is not the main focus of this manuscript, we can agree to disagree on this issue.

Point-by-point Response to Reviewers

Dear Reviewers:

We are really grateful for your approval for our revised manuscript. We also feel lucky that you provided valuable insights and thoughtful suggestions to guide our future research. Thank you very much for your dedications. Below you will find our point-by-point responses to the reviewers' comments:

Reviewer #1 (Remarks to the Author):

The authors have addressed my main concerns in this revision. It is a nice paper.

Response :

Thank you for your approval. Thank you very much for your dedication for reviewing our manuscript.

Reviewer #2 (Remarks to the Author):

Guo et al. satisfactorily addressed most of my concerns in this revision. The new display of data in the figures is more straightforward and easier to understand. The added statistic for Western blots builds confidence in their conclusions. One concern left remaining is that their data does not distinguish whether OsEPFL8 and OsEPFL9 proteins isolated from insect cells are correctly folded and suitable for ITC or whether OsER1 is not their main receptor. This refers to Fig 3c and d; item 4 in the previous review. The Authors discuss this point in the paper and in their rebuttal but in a rather confusing manner. The fact that OsEPFL6 isolated from insect cells is functional does not guarantee that other OsEPFLs isolated from insect cells are functional. Based on our experience working with small plant Cys -rich peptides isolated from insect cells, some peptides do not fold correctly, and as the result have low affinity for receptors and are physiologically inactive. Other peptides from the same gene family fold correctly and are functional in vivo. Based on the data in the manuscript it is not clear whether the OsEPFL8 and 9 peptides are of high quality and the data obtained in the

ITC experiment is reliable. However, since affinity of OsEPFL6,7,8,9 for OsER1 is not the main focus of this manuscript, we can agree to disagree on this issue.

Response :

Thank you very much for your profound comments and concern on this issue. In fact, we also considered this issue whether the lower affinity of OsEPFL8 and OsEPFL9 peptides is due to their features or whether OsEPFL8 and OsEPFL9 peptides isolated from insect cells are somewhat incorrectly folded. However, according to our experience, we tended to think that OsEPFL8 and OsEPFL9 peptides isolated from insect cells are correctly folded and suitable for assays *in vitro*. Certainly, we couldn't completely exclude the possibility that OsER1 is not their main receptor. Here, we appreciate you share your experience working with small plant Cys-rich peptides isolated from insect cells. We believe that it is possible that some small peptides couldn't be folded correctly and even are physiologically inactive, which results in lower affinity for receptors. In response to this, we have rephrased in the relevant discussion part (**described in the revised manuscript line 581**). We hope you are satisfied with our response. Thank you very much for your dedication for reviewing our manuscript.